# Extramacrochaetae regulates Notch signaling in the *Drosophila* eye through non-apoptotic caspase activity

Sudershana Nair[1], Nicholas E Baker[1,2,3]*[†, ‡]

[1]Department of Genetics, Albert Einstein College of Medicine, Bronx, United States; [2]Department of Developmental and Molecular Biology, Albert Einstein College of Medicine, Bronx, United States; [3]Department of Ophthalmology and Visual Sciences, Albert Einstein College of Medicine, Bronx, United States

*For correspondence:
nebaker@uci.edu

Present address: [†]Department of Neuroscience and Physiology, NYU School of Medicine, New York, United States; [‡]Department of Microbiology and Molecular Genetics, University of California, Irvine, United States

## eLife Assessment

This **important** work presents data showing that all non-proneural phenotypes of the Inhibitor of DNA binding (Id) protein Emc are mediated through inappropriate nonapoptotic caspase activity. Using the developing *Drosophila* retina as a model the authors show that Emc acts by transcriptionally regulating the *Death-Associated Inhibitor of Apoptosis 1* (*diap1*) gene, which impacts on Notch signaling by caspase-dependent increase of Delta protein. These are **compelling** findings, interesting for the caspase/apoptosis field as they add more non-apoptotic functions of caspases to the list, as well as for the Id field, which examines how Id proteins inhibit cell differentiation.

**Abstract** Many cell fate decisions are determined transcriptionally. Accordingly, some fate specification is prevented by Inhibitor of DNA-binding (Id) proteins that interfere with DNA binding by master regulatory transcription factors. We show that the *Drosophila* Id protein Extra macrochaetae (Emc) also affects developmental decisions by regulating caspase activity. Emc, which prevents proneural bHLH transcription factors from specifying neural cell fate, also prevents homodimerization of another bHLH protein, Daughterless (Da), and thereby maintains expression of the *Death-Associated Inhibitor of Apoptosis* (*diap1*) gene. Accordingly, we found that multiple effects of *emc* mutations on cell growth and on eye development were all caused by activation of caspases. These effects included acceleration of the morphogenetic furrow, failure of R7 photoreceptor cell specification, and delayed differentiation of non-neuronal cone cells. Within *emc* mutant clones, Notch signaling was elevated in the morphogenetic furrow, increasing morphogenetic furrow speed. This was associated with caspase-dependent increase in levels of Delta protein, the transmembrane ligand for Notch. Posterior to the morphogenetic furrow, elevated Delta cis-inhibited Notch signaling that was required for R7 specification and cone cell differentiation. Growth inhibition of *emc* mutant clones in wing imaginal discs also depended on caspases. Thus, *emc* mutations reveal the importance of restraining caspase activity even in non-apoptotic cells to prevent abnormal development, in the *Drosophila* eye through effects on Notch signaling.

## Introduction

Inhibitor of DNA-binding proteins (Id proteins), which are HLH proteins lacking basic DNA-binding sequences, form inactive heterodimers with proneural bHLH proteins, preventing DNA binding and transcriptional regulation of target genes (*Benezra et al., 1990*; *Cabrera et al., 1994*; *Ellis, 1994*; *Ellis et al., 1990*; *Garrell and Modolell, 1990*; *Norton, 2000*). Accordingly, mutations in Id protein

genes, of which there are four in mammals and one in *Drosophila*, permit enhanced proneural bHLH function, deregulating neurogenesis at many developmental stages and in many tissues (*Ling et al., 2014*; *Oproescu et al., 2021*; *Roschger and Cabrele, 2017*; *Wang and Baker, 2015a*).

Is this the only mechanism of Id protein function? Not all Id gene mutant phenotypes resemble gain of proneural gene function (*Ling et al., 2014*; *Wang and Baker, 2015a*). The sole *Drosophila* Id protein gene *extra macrochaetae* (*emc*) is required for normal growth of imaginal discs (*Alonso and Garcia-Bellido, 1988*). Imaginal discs are larval progenitors of adult tissues and remain proliferative and undifferentiated until late in the final larval instar; no proneural gene is active in these proliferating cells. A second example is that *emc* is required for ovarian follicle cell development (*Adam and Montell, 2004*), which does not depend on proneural bHLH genes. In *Drosophila* eye development, *emc* is required for the specification of the R7 photoreceptor cells and the non-neuronal cone cells, neither of which depends on any known proneural bHLH gene, and also restrains the rate at which the morphogenetic furrow, a wave of fate specification that traverses across the eye imaginal disc as retinal patterning and differentiation begin (*Bhattacharya and Baker, 2009*; *Brown et al., 1995*).

Insights into how Emc acts independently of proneural genes have emerged from several studies. Regarding the proper proliferation of imaginal disc cells (*Alonso and Garcia-Bellido, 1988*), independence of imaginal disc growth from proneural bHLH genes is confirmed by the lack of requirement for *da*, the only E protein in *Drosophila* (*Andrade-Zapata and Baonza, 2014*; *Bhattacharya and Baker, 2011*). E proteins are ubiquitously expressed bHLH proteins that are heterodimer partners for proneural bHLH proteins. Proneural bHLH proteins cannot function without Da, so normal growth of *da* mutant cells implies independence from all Da heterodimer partners. Instead, it is ectopic Da activity that causes poor growth in *emc* mutant cells, because normal growth is restored in *emc da* double mutant cells (*Bhattacharya and Baker, 2011*). Elevated Da levels activate transcription of *expanded* (*ex*), a component of the Salvador–Warts–Hippo (SWH) tumor suppressor pathway, to inhibit growth of imaginal disc cells (*Wang and Baker, 2015b*).

In the course of these studies, we noticed that *ex* mutations affected the number of sensory organ precursor (SOP) cells in the thorax (*Wang and Baker, 2015b*). This was due to activity of Yorkie (Yki), the target of SHW signals, in the *ex* mutant cells. Yki in turn increased transcription of a direct Yki target gene, *Death-Associated Inhibitor of Apoptosis 1* (*diap1*) (*Wang and Baker, 2019*). Diap1 is a ubiquitin ligase that prevents accumulation of a processed form of Dronc, an apical caspase in *Drosophila* (*Hawkins et al., 2000*; *Meier et al., 2000*). Caspases are proteases that drive apoptosis by cleaving cellular substrates. They are expressed as inactive zymogens that are then activated by signals (initiator caspases) or by cleavage by caspases (effector caspases) in a feed-forward process that is expected to kill cells rapidly once a threshold of caspase activity has been crossed (*Fuchs and Steller, 2011*). Caspases can also mediate non-apoptotic processes. What differentiates apoptotic from non-apoptotic outcomes is uncertain, as the same caspase cascade of initiator and effector caspases seems to be involved in both (*Aram et al., 2017*; *Baena-Lopez et al., 2018*; *Colon-Plaza and Su, 2022*; *Kanuka et al., 2005*; *Kuranaga and Miura, 2007*; *Nakajima and Kuranaga, 2017*; *Su, 2020*). In *ex* mutants, the increased *diap1* transcription does not affect cell death much but does cause an increase in *wg* signaling activity (*Wang and Baker, 2019*). Wg signaling promotes SOP cell determination in the thorax, and it has been shown previously that Wg signaling is antagonized by caspases, through a mechanism that is non-apoptotic but whose direct target is not yet certain (*Baker, 1988*; *Kanuka et al., 2005*; *Wang and Baker, 2019*).

Since *emc* mutations elevate *ex* expression (*Wang and Baker, 2015b*), we wondered whether *emc* also changes Diap1 levels and affects caspase activities. Increased *ex* expression would be expected to reduce transcription of the *diap1* gene and potentially increase caspase activity, the opposite of what occurs in *ex* mutants. Remarkably, we found that caspase activity was responsible for all the aspects of the *emc* phenotype that we tested. This included reduced growth of imaginal disc cells, accelerated morphogenetic furrow progression, and loss of R7 and cone cells fates in the eye. The effects on eye development all reflected caspase-dependent expression of the Notch ligand Delta, which is known to contribute to morphogenetic furrow progression, and to R7 and cone cell fate specification. Thus, caspase-dependent non-apoptotic signaling underlies multiple roles of *emc* that are independent of proneural bHLH proteins.

## Results

### Caspase activity causes growth defects in *emc* mutants

To determine the contribution of caspases to growth inhibition in *emc* mutant cells, we used the Flippase (FLP)/FLP Recombinase Target (FRT) system (*Xu and Rubin, 1993*) to generate clones of *emc* mutant cells that were unable to activate caspases normally. We achieved this by homozygously deleting the linked *reaper* (*rpr*), *head involution defective* (*hid*), and *grim* genes using *Df(3L)H99*. Rpr, Hid, and Grim proteins promote Diap1 degradation in response to apoptotic stimuli and allow activation of initiator caspases such as *dronc* to start apoptosis (*Yoo et al., 2002*). By removing all three pro-apoptotic proteins, *Df(3L)H99* affects both apoptotic and non-apoptotic caspase functions in *Drosophila* (*Tapadia and Gautam, 2011*; *White et al., 1994*). Thus, clones of *emc Df(3L)H99* mutant cells should be defective for caspase activation. In addition, we also made clones of *emc* mutant cells also mutated for *dronc*, which encodes the main initiator caspase in *Drosophila* that is necessary for most developmental apoptosis (*Hawkins et al., 2000*; *Meier et al., 2000*; *Quinn et al., 2000*). Dronc contributes to non-apoptotic caspase functions as well, so *emc dronc* mutant cells should also show reduced non-apoptotic and well as apoptotic caspase functions.

In comparison to neutral clones, which grew equivalently to their twin-spot controls, *emc* mutant clones showed greatly reduced growth in eye or wing imaginal discs, as described previously (*Figure 1A–D*, *Andrade-Zapata and Baonza, 2014*; *Bhattacharya and Baker, 2011*; *Alonso and Garcia-Bellido, 1988*). In contrast, *emc* mutant clones that were also mutant for *dronc* (*dronc*[i29]; *Xu et al., 2005*), or *emc Df(3L)H99* clones that lacked the *rpr*, *grim*, and *hid* genes, grew more like their twin-spot controls in both eye and wing (*Figure 1E–H*). Statistically, growth of *emc* clones also deleted for the *rpr*, *hid*, and *grim* genes was not distinguishable from the wild-type, and similarly for *emc* clones also mutated for *dronc* (see *Figure 1N* for quantification). Neither homozygosity for *dronc*, nor deletion of *rpr*, *grim*, and *hid*, significantly affected growth of otherwise wild-type imaginal disc clones (*Figure 1I–L*).

Previous studies of *emc* phenotypes have often used Minute genetic backgrounds (i.e. heterozygosity for mutations in *Rp* genes) to retard growth and enhance the size of the *emc* mutant clones (*Bhattacharya and Baker, 2009*). When *emc Df(3L)H99* clones were induced in the *M(3)67C* background, the clones took over almost the entire disc, leaving only a few *M(3)67C* heterozygous cells remaining (*Figure 1M*). The growth advantage of *emc dronc* clones was not so marked.

These results indicate that the growth disadvantage of *emc* mutant imaginal disc clones is mostly attributable to cell death genes. Preventing caspase activation, either by mutating the main initiator caspase, or by preventing Diap1 turnover, partially or completed restored normal growth. Our data did not support a previous suggestion that *dronc* was required for normal wing disc growth (*Verghese et al., 2012*).

### Emc regulates furrow progression through non-apoptotic caspase activity

*Emc* mutations have multiple effects on the eye imaginal disc, although only a few aspects of retinal differentiation depend on proneural bHLH genes. The proneural gene *atonal* is required, along with *da*, for the specification of R8 photoreceptor precursors in the morphogenetic furrow that initiate each ommatidial cluster in the larval eye disc (*Brown et al., 1996*; *Jarman et al., 1994*). Later, during pupal development, proneural genes of the Achaete-Scute gene Complex (AS-C), along with *da*, are required for the specification of the interommatidial bristles (*Cadigan et al., 2002*). All the other cell types develop independently of proneural bHLH genes, and most of them develop independently of *da* (*Brown et al., 1996*; *Jiménez and Campos-Ortega, 1987*).

In *emc* mutant clones, retinal differentiation begins precociously, associated with more rapid transit of the morphogenetic furrow across the disc (*Figure 2A, B*, *Bhattacharya and Baker, 2011*; *Bhattacharya and Baker, 2012*; *Brown et al., 1995*). In contrast, we found that 75% of the time, the morphogenetic furrow progressed normally through *emc dronc* double mutant clones (*Figure 2C*). Eye discs containing *emcDf(3L)H99* double mutant clones always appeared completely normal (*Figure 2D*). Control *dronc* and *Df(3L)H99* mutant clones that lacked *emc* mutations also showed normal furrow progression (*Figure 2—figure supplement 1A, B*).

In addition, *emc* mutant clones also exhibit sporadic ectopic differentiation of neurons anterior to the morphogenetic furrow, which do not take photoreceptor cell fate (*Figure 2A, B*; *Bhattacharya*

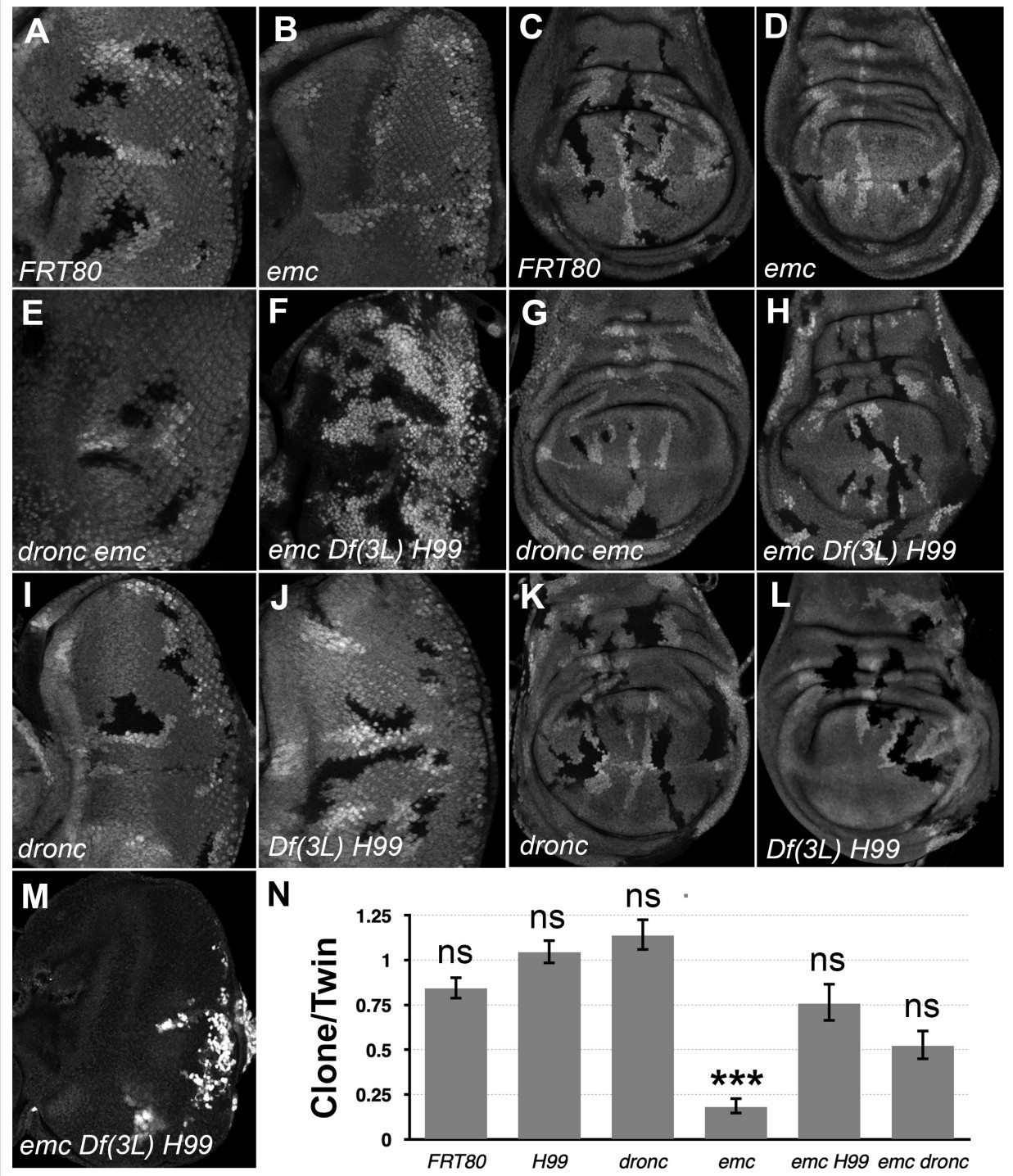

**Figure 1.** *Df(3L)H99* recues growth defect of *emc* mutant clones. (**A–L**) Control and mutant clones in eye and wing imaginal disc were induced at the end of first instar and are associated with sibling twin-spot clones marked by two copies of the GFP marker (brighter gray) that serves as internal control for growth. Clones are labeled by the absence of GFP. (**M**) *emc H99* clones induced in the Minute background (**N**) Quantification of the ratio of clone size to twin-spot measured in wing imaginal discs. Geometrical means ± SEM are shown. After log-transformation of clone/twin-spot ratios to ensure normality, one-way ANOVA rejected the null hypothesis that these results are the same ($p = 5.72 \times 10^{-8}$). The Holm correction for multiple comparison was used to identify significant differences between all pairs of samples. *** denotes highly significant difference from the FRT80 control ($p < 0.001$), NS denotes no significant difference ($p > 0.05$). Whereas the clone/twin-spot ratio for *emc* homozygous clones was significantly different from the FRT80 control ($p = 5.72 \times 10^{-6}$), this was not true for any of the other genotypes (*H99*, $p = 0.79$; *dronc*, $p = 0.906$; *emc H99*, $p = 0.92$; *emc dronc*, $p = 0.345$). The clone/twin-spot ratio for *emc* homozygous clones was also significantly different from that for *emc H99* or *emc dronc* ($p = 4.12 \times 10^{-6}$ and $p =$

*Figure 1 continued on next page*

*Figure 1 continued*

0.0177, respectively), whereas *emc H99* and *emc dronc* did not differ significantly from *H99* or *dronc* clones (p = 1 and p = 0.136, respectively). Source data for (**N**) are provided in *Figure 1—source data 1*. Genotypes: (**A, C**) *ywhsF;FRT80/[UbiGFP]FRT80*, (**B, D**) *ywhsF;emc^{AP6}FRT80/[Ubi-GFP]FRT80*, (**E, G**) *ywhsF;dronc^{i29}emc^{AP6} FRT80/[UbiGFP]FRT80*, (**F, H**) *ywhsF;emc^{AP6} Df(3L)H99 FRT80/[UbiGFP]FRT80*, (**I, K**) *ywhsF;dronc^{i29} FRT80/[UbiGFP]FRT80*, (**J, L**) *ywhsF;;Df(3L)H99FRT80/[UbiGFP]FRT80*, (**M**) *ywhsF; emc^{AP6} Df(3L)H99 FRT80/[UbiGFP] M(3)67C FRT80*. N = 10 for each genotype.

The online version of this article includes the following source data for figure 1:

**Source data 1.** Clone size source data for *Figure 1N*.

*and Baker, 2011*; *Brown et al., 1995*). This was likewise absent from *emc dronc* and *emcDf(3L)H99* double mutant clones (*Figure 2C, D*).

The overall pattern of retinal differentiation revealed by labeling for Senseless, which is specific for R8 photoreceptor cells in the retina, and Elav, which labels all neuronal photoreceptor cells, appeared so normal in these genotypes that we deemed it necessary to confirm that the supposedly *emc dronc* mutant and *emc Df(3L)H99* mutant clones were indeed mutated for *emc*. This was confirmed by lack of staining by an antibody against Emc protein, similar to plain *emc* mutant clones that did affect the morphogenetic furrow (*Figure 2—figure supplement 1C–E*). We also found that *emc dronc* and *emc Df(3L)H99* clones upregulated Da protein expression to a comparable degree to plain *emc* mutant clones (*Figure 2—figure supplement 1C–E*, *Bhattacharya and Baker, 2011*). This further confirmed the absence of *emc* function from *emc dronc* mutant and *emc Df(3L)H99* genotypes, and also showed that cell death pathways were not required for the regulation of Da protein levels by *emc*.

We have previously concluded that *diap1* transcription is reduced in *emc* mutant clones, and in Da-overexpressing cells, due to Yki inhibition downstream of *ex* (*Wang and Baker, 2015b*). In eye imaginal discs, Diap1 protein was normally present uniformly (*Figure 3A*). Diap1 protein levels were cell-autonomously reduced in *emc* mutant clones posterior to the morphogenetic furrow, compared to *emc/+* cells in the same eye discs, and shown and quantified in (*Figure 3B, E*). A comparable reduction was also seen in *emc Df(3L)H99* clones, compared to *emc/+ Df(3L)H99/+* cells in the same eye discs (*Figure 3C, E*). There was also no difference in Diap1 levels between *Df(3L)H99* homozygous clones and *Df(3L)H99/+* cells in the same eye discs (*Figure 3D, E*). These data suggest emc affects Diap1 protein levels, whereas rpr, grim, and hid affect Diap1 protein activity. A caveat is that Diap1 levels in mutant clones were not compared directly to wild-type cells, none of which were present in the same eye discs, and that *Df(3L)H99* might affect Diap1 protein levels dominantly. Interestingly, we did not detect *emc*-dependent changes in Diap1 levels in wing discs (*Figure 3—figure supplement 1B*).

To test the role of Diap1 in the eye directly, *diap1* was over-expressed in the posterior eye by transcription under GMR-Gal4 control. This rescued morphogenetic furrow speed to normal in *emc* clones (*Figure 3—figure supplement 2B*). We also saw restoration of morphogenetic furrow speed in *emc* clones that expressed Baculovirus P35 under GMR-Gal4 control (*Figure 3—figure supplement 2A*). Baculovirus P35 encodes a caspase pseudo-substrate that inhibits all *Drosophila* caspases except Dronc (*Hawkins et al., 2000*; *Hay et al., 1994*; *Meier et al., 2000*; *Xue and Horvitz, 1995*). The *emc* clones expressing p35 also lacked cell death (*Figure 3—figure supplement 3B*). The restoration of morphogenetic furrow speed by Diap1, a Dronc antagonist, as well as by the caspase inhibitor P35, suggest that the effect of morphogenetic furrow acceleration in *emc* mutant clones is due to the caspase cascade.

To test whether reduced Diap1 expression promoted apoptosis and thereby accelerated the morphogenetic furrow, we assessed apoptosis levels by terminal deoxynucleotidyl transferase dUTP nick end labeling (TUNEL) of eye discs containing mutant clones. We found almost no cell death posterior to the furrow in *emc* clones (*Figure 3—figure supplement 2D*), and cell death in *emc* clones anterior to the furrow was comparable to controls and less than that of cells surrounding the clones (*Figure 3—figure supplement 2C, D*). Some cell death is expected outside *emc* or control clones, as these cells have the *M* background that is itself associated with an increase in apoptosis (*Coelho et al., 2005*; *Kale et al., 2015*; *Li and Baker, 2007*). Similar to TUNEL, we found no cell death in *emc* clones posterior the furrow with Dcp1 staining (*Figure 3—figure supplement 3D*). To test whether apoptosis in the eye disc would be sufficient to promote morphogenetic furrow progression, we generated mosaic clones for a *pineapple eye* (*pie*) mutation. These *pie* mutant cells have an elevated rate of apoptosis in imaginal discs, but not sufficient to prevent *pie* homozygous clones surviving late

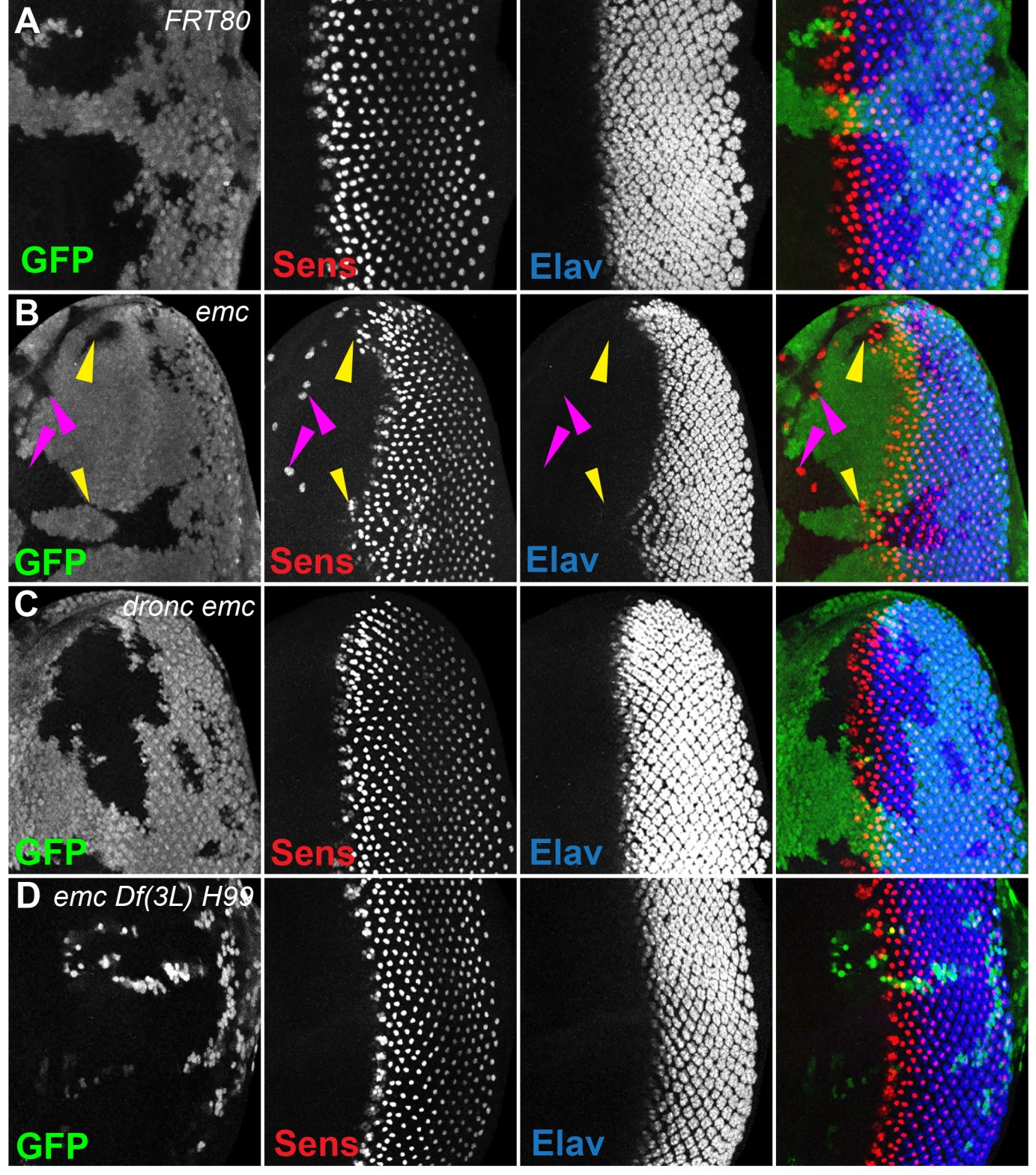

**Figure 2.** Morphogenetic furrow progression is affected by cell death pathways. In all panels, the differentiating neurons are marked by Elav (in blue), and mutant cells are identified by the absence of GFP expression (in green). (**A**) The wave of retinal differentiation from posterior to anterior (right to left), marked here by Senseless expression in R8 photoreceptor cells (red) is normal in FRT80 control clones. (**B**) *emc* null clones lacking GFP show acceleration of retinal differentiation illustrated by yellow arrows (premature differentiation can also continue into wild-type regions ahead of such clones). In addition, ectopic neural differentiation also occurs sporadically anterior to the morphogenetic furrow, and unassociated with it (magenta

*Figure 2 continued on next page*

*Figure 2 continued*

arrows). (**C**) In contrast, retinal differentiation proceeds at the same pace in *emc dronc* double mutant clones as in nearby wild-type regions in 75% of the eye discs. (**D**) *emc H99* double mutants show a stronger suppression of the acceleration of retinal differentiation (compare panel B). Genotypes: (**A**) *ywhsF;FRT80/[UbiGFP] M(3)67C FRT80*, (**B**) *ywhsF;emc^{AP6}FRT80/[Ubi-GFP] M(3)67C FRT80*, (**C**) *ywhsF;dronc^{i29}emc^{AP6} FRT80/[UbiGFP] M(3)67C FRT80* (*n* = 12), (**D**) *ywhsF;emc^{AP6} Df(3L)H99 FRT80/[UbiGFP] M(3)67C FRT80*. N = 8 for each genotype.

The online version of this article includes the following figure supplement(s) for figure 2:

**Figure supplement 1.** Morphogenetic furrow progression in *emc* mutant cells.

into larval and even adult life (*Shi et al., 2003*). The rate of morphogenetic furrow progression was unaffected in eye discs containing *pie* clones (*Figure 3—figure supplement 3A*). Because no excess cell death was detected in *emc* clones, and cell death was insufficient to accelerate the morphogenetic furrow in otherwise wild-type eye discs, *emc* clones might be affected by a non-apoptotic caspase activity.

## Wingless and Dpp signaling are unaffected by *emc*

To understand how caspases could affect the speed of the morphogenetic furrow, we analyzed pathways known to contribute. Hedgehog (Hh) and Decapentaplegic (Dpp) signaling drive this differentiation wave, along with a contribution from Notch signaling (*Baonza and Freeman, 2001*; *Borod and Heberlein, 1998*; *Fu and Baker, 2003*; *Heberlein et al., 1993*; *Ma et al., 1993*). A negative regulator of morphogenetic furrow progression is Wingless (Wg), which is expressed at the dorsal and ventral eye disc margins (*Lee and Treisman, 2002*; *Maurel-Zaffran and Treisman, 2000*).

Because we found that *ex* mutations affected thoracic bristle patterning through a caspase-dependent non-apoptotic effect on Wg signaling (*Wang and Baker, 2019*), we looked first to see whether *emc* mutations reduced Wg signaling in the eye. We used Frizzled-3 RFP (Fz3-RFP) as a reporter (*Sato et al., 1999*). In control eye discs, the Fz3-RFP recapitulates the pattern of endogenous Wg signaling activity at the wing margins (*Figure 4—figure supplement 1*, *Treisman and Rubin, 1995*). Frizzled-3 RFP expression was not changed in *emc* clones (*Figure 4A*). We then used a mutation in *naked cuticle* (*nkd*), encoding a negative feedback regulator of Wg signaling, to modulate Wg signaling (*Chang et al., 2008*; *Zeng et al., 2000*). If the morphogenetic furrow was accelerated in *emc* mutant clones due to reduced Wg signaling, more normal development should occur in *emc nkd* clones. The morphogenetic furrow was still accelerated in *emc nkd* clones, however (*Figure 4B*). These results provided no evidence that Wg signaling was the relevant *emc* target in the eye.

We also examined Dpp signaling, since ectopic Dpp signaling is sufficient to accelerate the morphogenetic furrow (*Pignoni and Zipursky, 1997*). The pattern of pMad, a readout of Dpp signaling, was identical in *emc* mutant and control clones spanning the morphogenetic furrow (*Figure 4C, D*). Thus, Dpp signaling did not seem altered by *emc* mutants either.

## Hedgehog pathway

Hedgehog (Hh) signaling is a key mover of the morphogenetic furrow (*Heberlein et al., 1993*; *Ma et al., 1993*; *Treisman, 2013*). Elevated Hh signaling is sufficient to accelerate the morphogenetic furrow (*Heberlein et al., 1995*; *Ma and Moses, 1995*). Notably, it has been suggested previously that *emc* mutations affect the morphogenetic furrow by activating Hedgehog signaling, because *emc* mutant cells accumulate Ci protein (*Spratford and Kumar, 2013*). Full-length Ci protein (Ci155) is targeted to the proteosome by Cul1 for processing into a transcriptional repressor protein Ci75 (*Aza-Blanc et al., 1997*). By inhibiting this processing, Hh prevents repression of target genes by Ci75 and promotes transcriptional activation downstream of Ci155. Accordingly, Ci155 accumulation is a feature of cells receiving Hh signals (*Motzny and Holmgren, 1995*).

We confirmed that *emc* mutant cells contain higher levels of Ci155, as reported previously (*Figure 5A*, *Spratford and Kumar, 2013*). Ci155 was elevated in *emc dronc* clones (*Figure 5B*) but reduced to wild-type levels in *emc Df(3L)H99* clones (*Figure 5C*, *Figure 5—figure supplement 1*). Thus, Ci155 levels did not correlate perfectly with behavior of the morphogenetic furrow.

We noticed that Ci155 levels were elevated in *emc* mutant clones in the posterior, differentiating eye disc, as well as in and ahead of the morphogenetic furrow (*Figure 5A*). This is significant, because Ci155 is not affected by Hh-dependent Cul1 processing posterior to the morphogenetic furrow (*Baker*

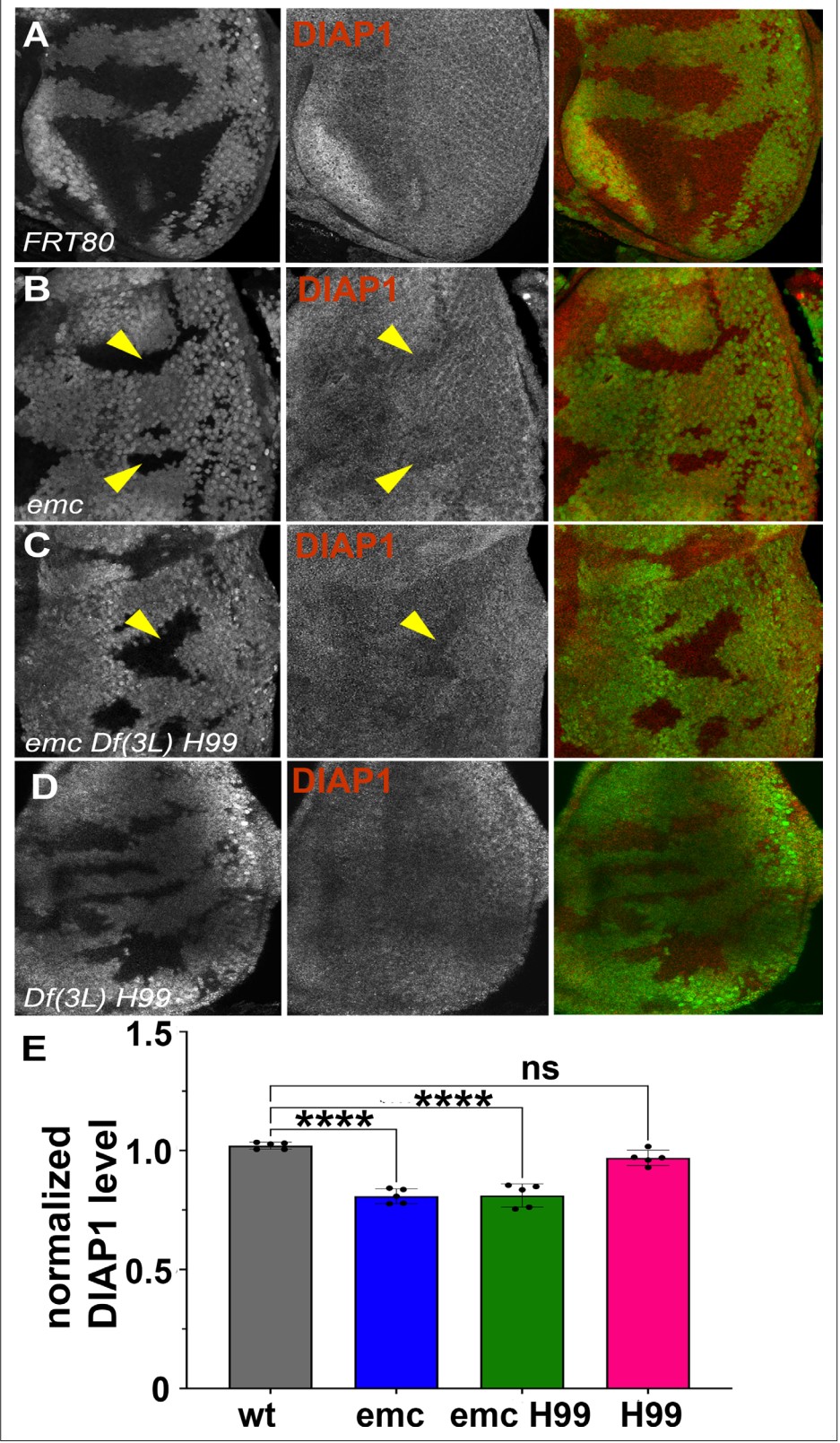

**Figure 3.** Diap1 expression in *emc* clones. In mosaic eye discs with (**A**) FRT80 clones, there is no difference in Diap1 levels within and outside the clone. However, both (**B**) *emc* mutant clones and (**C**) *emc H99* clones show reduced Diap1 levels posterior to the morphogenetic furrow, compared to the heterozygous background. (**D**) *H99* clones showed similar Diap1 levels, compared to the heterozygous background. (**E**) Quantification of

*Figure 3 continued on next page*

*Figure 3 continued*

Diap1 levels in different clone genotypes, compared to the background levels outside the clones. Means ± SEM are shown. Note that our experiments did not generate mosaics of wild-type and Df(3L)H99/+ cells for direct comparison of these genotypes. Statistical significance calculated by two-way ANOVA (****$p \leq 0.0001$). Source data for (**E**) are provided in *Figure 3—source data 1*. Genotypes: (**A**) *ywhsF;FRT80/[UbiGFP] M(3)67C FRT80*, (**B**) *ywhsF;emc^AP6^FRT80/[Ubi-GFP] M(3)67C FRT80*, (**C**) *ywhsF;emc^AP6^ Df(3L)H99 FRT80/[UbiGFP] M(3)67C FRT80*, (**D**) *ywhsF; Df(3L)H99 FRT80/[UbiGFP] M(3)67C FRT80*. N = 6 for each genotype.

The online version of this article includes the following source data and figure supplement(s) for figure 3:

**Source data 1.** Anti-Diap1 labeling source data for *Figure 3E*.

**Figure supplement 1.** DIAP1 staining in wing disc clones.

**Figure supplement 2.** Apoptosis and furrow progression.

**Figure supplement 3.** Emc clones show no caspase activity beyond the morphogenetic furrow.

*et al., 2009*; *Ou et al., 2002*). Ci155 accumulation posterior to the furrow suggests a Hh-independent mechanism.

To test whether Ci155 accumulation in *emc* clones indicates elevated Hh target signaling, we looked at the transmembrane receptor Patched (Ptc) which is a transcriptional target of Hh signaling, acting in a negative feedback loop (*Hepker et al., 1997*). We saw no changes in Ptc protein levels in either *emc* or *emc Df(3L)H99* clones compared to controls, questioning the notion that Hh signaling was altered by *emc* (*Figure 5—figure supplement 2*).

To test functionally whether Ci155 is responsible for accelerating the furrow in *emc* clones, we generated *emc ci* double mutant clones. To achieve this, a genomic transgene that rescues *ci^94^* flies to adulthood (*Little et al., 2020*), was introduced into chromosome arm 3L where it is linked to the wild-type *emc* locus, so that mitotic recombination in the *ci^94^* null background leads to *emc ci* double mutant clones. For unknown reasons, *emc ci* double mutant clones were small and difficult to obtain, even in the Minute background. Eye differentiation was still accelerated in those *emc ci* double mutant clones we found that spanned the morphogenetic furrow (*Figure 5D*). The unexplained synergistic growth effects in particular could be consistent with interactions between *emc* and Hh signaling, but *emc* must regulate the speed of the morphogenetic furrow through at least one other target besides Ci in order to explain the furrow acceleration observed in *emc ci* double mutant clones.

## Delta expression is a target of caspases

The remaining signaling pathway that contributes to morphogenetic furrow movement is Notch. Specifically, only cells where Notch signaling is active are competent to initiate retinal differentiation in response to Dpp (*Baonza and Freeman, 2001*; *Fu and Baker, 2003*). Accordingly, ectopic expression of Delta, the transmembrane ligand for Notch, is sufficient to accelerate retinal differentiation anterior to the morphogenetic furrow by expanding the effective range of Dpp signaling (*Baonza and Freeman, 2001*; *Li and Baker, 2001*).

To test whether *emc* restrains Notch, we examined the bHLH proteins of the E(spl)-C, widely characterized targets of the Notch pathway (*Bray, 2006*). We used mAb323 to detect up to five E(spl) bHLH proteins with Notch-dependent expression (*Jennings et al., 1994*). E(spl) protein expression, and hence Notch signaling, was higher in the morphogenetic furrow in *emc* clones than in wild-type cells (*Figure 6A*). In contrast, E(spl) protein expression was normal in *emc dronc* clones (*Figure 6B*).

To test whether elevated Notch signaling was required to accelerate the morphogenetic furrow in *emc* mutants, we examined *emc psn* double mutant clones. Presenilin (Psn) is the enzymatic component of γ-secretase that releases the intracellular domain of Notch during active Notch signaling, and the *psn* gene is linked to *emc*. Loss of *psn* function leads to a Notch loss of function phenotype (*Struhl and Greenwald, 1999*; *Ye et al., 1999*). Accordingly, *psn* clones lead to a neurogenic phenotype in the eye, without affecting the progression of the morphogenetic furrow (*Figure 6C*, *Li and Baker, 2001*). The position of the morphogenetic furrow was also not affected in *emc psn* clones (*Figure 6D*). Thus, the furrow was not accelerated in *emc* mutant clones also defective for Notch signaling.

These two results together indicated that *emc* mutants promoted Notch signaling in the morphogenetic furrow, acting through caspase signaling on a step prior to γ-secretase cleavage of the intracellular domain of Notch. Accordingly, we decided to check Delta (Dl) protein levels. We found that Dl

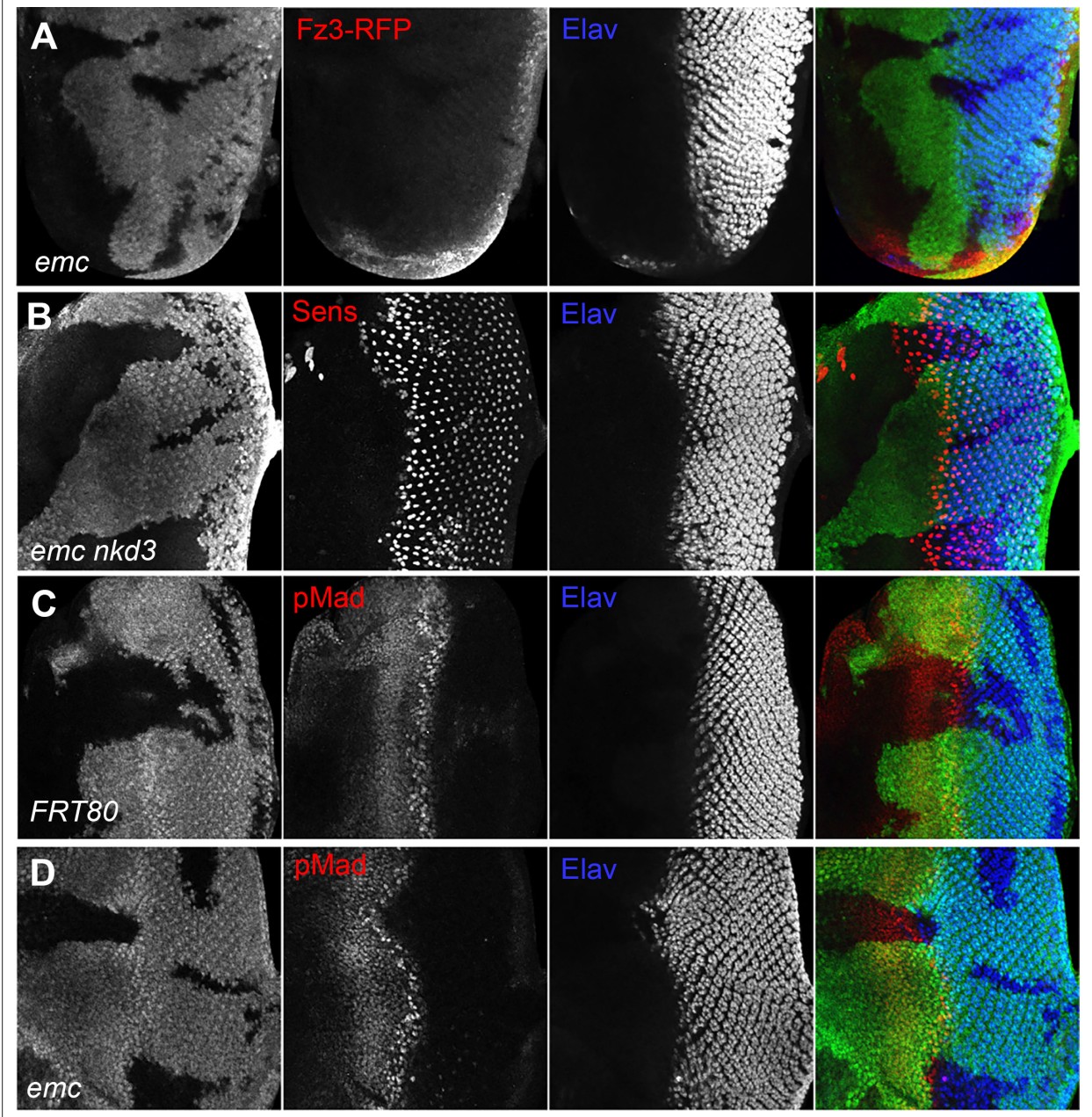

**Figure 4.** Wingless and Dpp signaling are not caspase targets. (**A**) No reduction in the Wg signaling reporter Fz3-RFP was detectable in emc clones. See **Figure 4—figure supplement 1** for Fz3-RFP expression in the wild-type. (**B**) Retinal differentiation is accelerated in *emc nkd³* double mutant clones like in *emc* clones. Senseless expression in R8 photoreceptor cells and Elav staining in differentiating photoreceptors are shown. (**C**) p-Mad accumulates around the morphogenetic furrow in eye discs containing control clones (**Firth et al., 2010**; **Vrailas and Moses, 2006**). (**E**) Except for the advanced progression, p-Mad levels were unchanged in *emc* clones. Genotypes: (**A**) *Fz3-RFP/+; emc^{AP6}FRT80/[Ubi-GFP] M(3)67C FRT80*, (**B**) *ywhsF;emc^{AP6}nkd³FRT80/[Ubi-GFP] M(3)67C FRT80*, (**C**) *ywhsF;FRT80/[UbiGFP] M(3)67C FRT80*, (**D**) *ywhsF;emc^{AP6}FRT80/[Ubi-GFP] M(3)67C FRT80. N* = 8 for each genotype.

The online version of this article includes the following figure supplement(s) for figure 4:

**Figure supplement 1.** Fz3-RFP/+ eye disc showing RFP and Elav staining.

protein levels were significantly and consistently elevated cell-autonomously throughout *emc* clones, both posterior and anterior to the furrow (**Figure 7B, E**). This included the region just ahead of the morphogenetic furrow that lacks Delta expression in normal development (**Baker and Yu, 1998**; **Parks et al., 1995**, **Figure 7B**). In contrast, levels of Dl protein in *emc Df(3L)H99* clones were similar to those

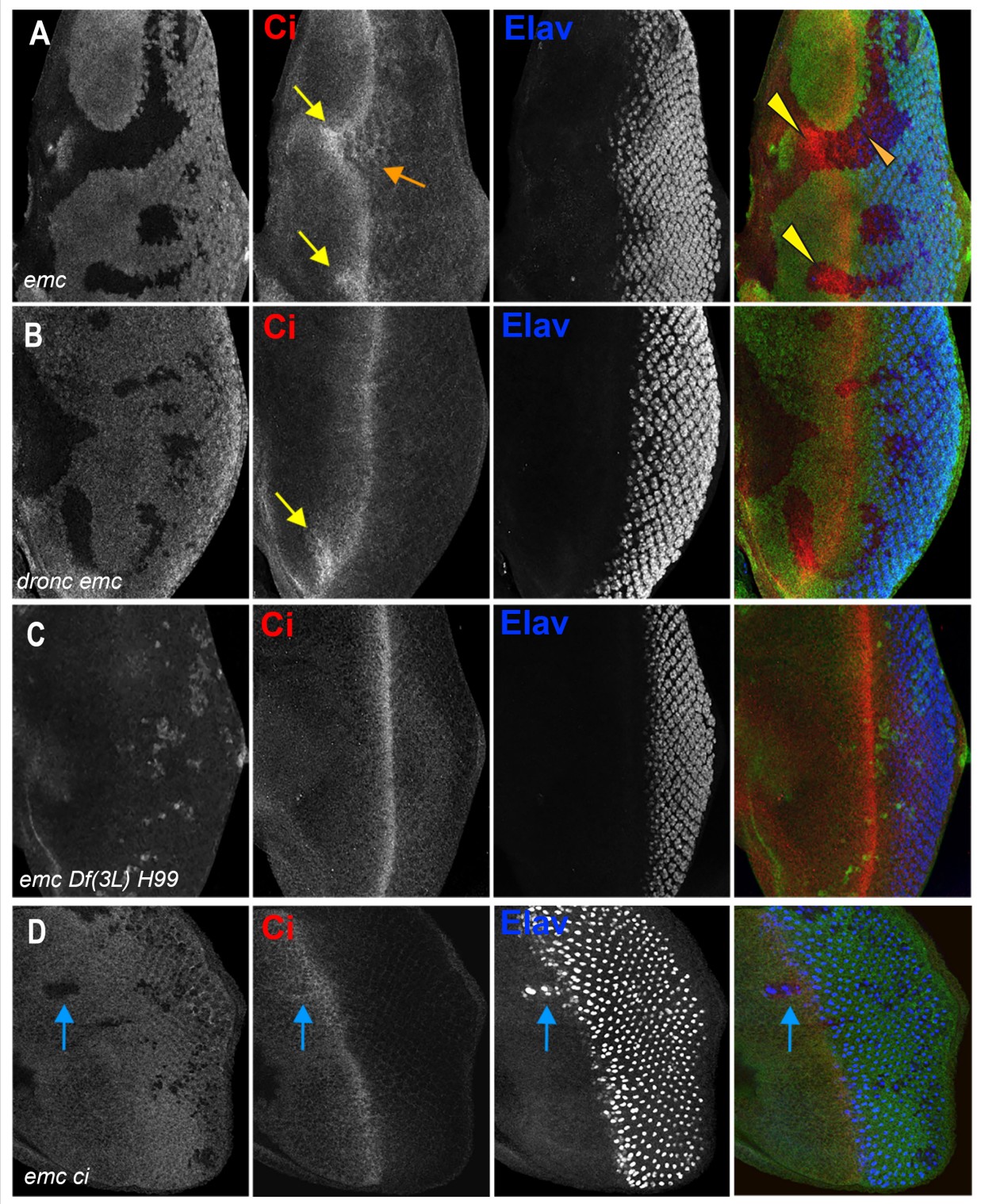

**Figure 5.** Ci expression and function in *emc* mutant clones. (**A**) Ci is elevated within *emc* mutant cells (yellow arrows). This was also true posterior to the morphogenetic furrow (orange arrow). (**B**) Higher Ci was also seen in *emc dronc* mutant cells, even when the morphogenetic furrow progressed normally, as indicated by Elav staining. (**C**) Ci levels were completely normal in *emc H99* clones. (**D**) *emc* and *ci* double mutant clones lacking GFP shows accelerated retinal differentiation (blue arrow). Genotypes: (**A**) *ywhsF;emc^{AP6}FRT80/[Ubi-GFP] M(3)67C FRT80*, (**B**) *ywhsF;dronc^{i29}emc^{AP6} FRT80/[UbiGFP] M(3)67C FRT80*, (**C**) *ywhsF;emc^{AP6} Df(3L)H99 FRT80/[UbiGFP] M(3)67C FRT80*, (**D**) *ywhsF;emc^{AP6}FRT80/[Ubi-GFP] ci^+ M(3)67C FRT80;ci[94]/ci[94]*. N = 8 for each genotype except (**D**) which has *n* = 3.

*Figure 5 continued on next page*

*Figure 5 continued*

The online version of this article includes the following figure supplement(s) for figure 5:

**Figure supplement 1.** Ci155 levels when cell death pathways are blocked.

**Figure supplement 2.** Patched staining in emc mutants.

of wild-type controls (*Figure 7A, D, E*). Levels in *emc dronc clones* were also similar to wild-type on average, although some clones seemed to show an increase, smaller than in *emc* clones (*Figure 7A, C, E*). This indicated that Dl protein is a target of caspases, directly or indirectly. The Delta protein sequence contains multiple predicted caspase target sites, as do many proteins (*Wang et al., 2014*). Only one candidate site lies in the intracellular domain, where it would potentially be accessible to caspases (*Figure 7—figure supplement 1*). Truncation of the Dl intracellular domain is usually associated with loss of Dl function, not stabilization and enhanced function, however (*Daskalaki et al., 2011*).

Because Dl activates Notch signaling cell non-autonomously, we wondered whether the effect of emc mutant clones was cell-autonomous. We note that, in all the experiments reported here, and in all the previous studies of emc mutant clones affecting morphogenetic furrow movement, the morphogenetic furrow is maintained as a continuous groove across the eye disc (*Bhattacharya and Baker, 2011*; *Bhattacharya and Baker, 2012*; *Brown et al., 1995*; *Spratford and Kumar, 2013*). That is, the advanced front of retinal differentiation within emc clones is always smoothly continuous with the normal morphogenetic furrow outside the clones, implying a progressive gradual increase in morphogenetic furrow speed near the lateral edges of emc mutant clones (e.g. *Figures 2B, 4B, D, 5A, and 6A*). Clones of *Su(H)* null mutants, which act cell-autonomously, provide a contrasting example. Complete loss of *Su(H)* accelerates the morphogenetic furrow due to loss of default Su(H) repression of Notch targets (*Li and Baker, 2001*). The boundaries of *Su(H)* null clones exhibit a clear discontinuity between the rates of differentiation within and outside the clones (*Figure 6E*). This difference between *Su(H)* and emc mutant clones is consistent with the idea that emc affects morphogenetic furrow progression differently from *Su(H)*.

## Specific ommatidial cell fates are regulated by caspases in *emc* mutants

Because the general pattern of neurogenesis revealed by pan-neuronal anti-Elav staining appeared so normal in *emc dronc* and *emc Df(3L)H99* clones (*Figure 2C, D*), we examined whether effects of *emc* on particular retinal cell fates was caspase dependent. Emc is also required for R7 differentiation and for timely onset of cone cell differentiation (*Bhattacharya and Baker, 2009*), two cell fate decisions that also depend on Notch signaling (*Cooper and Bray, 2000*; *Flores et al., 2000*; *Tomlinson and Struhl, 2001*; *Treisman, 2013*). These cell fates are normally independent of *da*, but like imaginal disc cell growth and morphogenetic furrow progression, ectopic *da* activity perturbs them in *emc* mutants (*Brown et al., 1996*; *Reddy Onteddu et al., 2024*).

Clones of *emc* mutant cells lack R7 photoreceptor cells (*Figure 8A, B*, *Bhattacharya and Baker, 2009*). R7 differentiation was restored to 90% of ommatidia in *emc Df(3L)H99* clones (*Figure 8C*). As shown before, *emc* mutant clones delayed cone cell differentiation by two to three columns (corresponding to a delay of 3–6 hr) (*Bhattacharya and Baker, 2009*, *Figure 8D, E*). We found no delay in cone cell differentiation in most *emc Df(3L)H99* clones (*Figure 8F*). These results indicated that caspase activity contributes to the R7 and cone cell differentiation defects that are also characteristic of *emc* mutants.

## Discussion

The Id proteins are important regulators of differentiation (*Ling et al., 2014*; *Massari and Murre, 2000*; *Roschger and Cabrele, 2017*; *Singh et al., 2022*; *Wang and Baker, 2015a*). They are well known to antagonize proneural basic helix–loop–helix (bHLH) proteins (*Benezra et al., 1990*; *Cabrera et al., 1994*; *Ellis, 1994*; *Ellis et al., 1990*; *Garrell and Modolell, 1990*; *Norton, 2000*). It has been uncertain whether this is their only function, as in *Drosophila* it is clear that *emc* mutations affect processes that are independent of proneural genes.

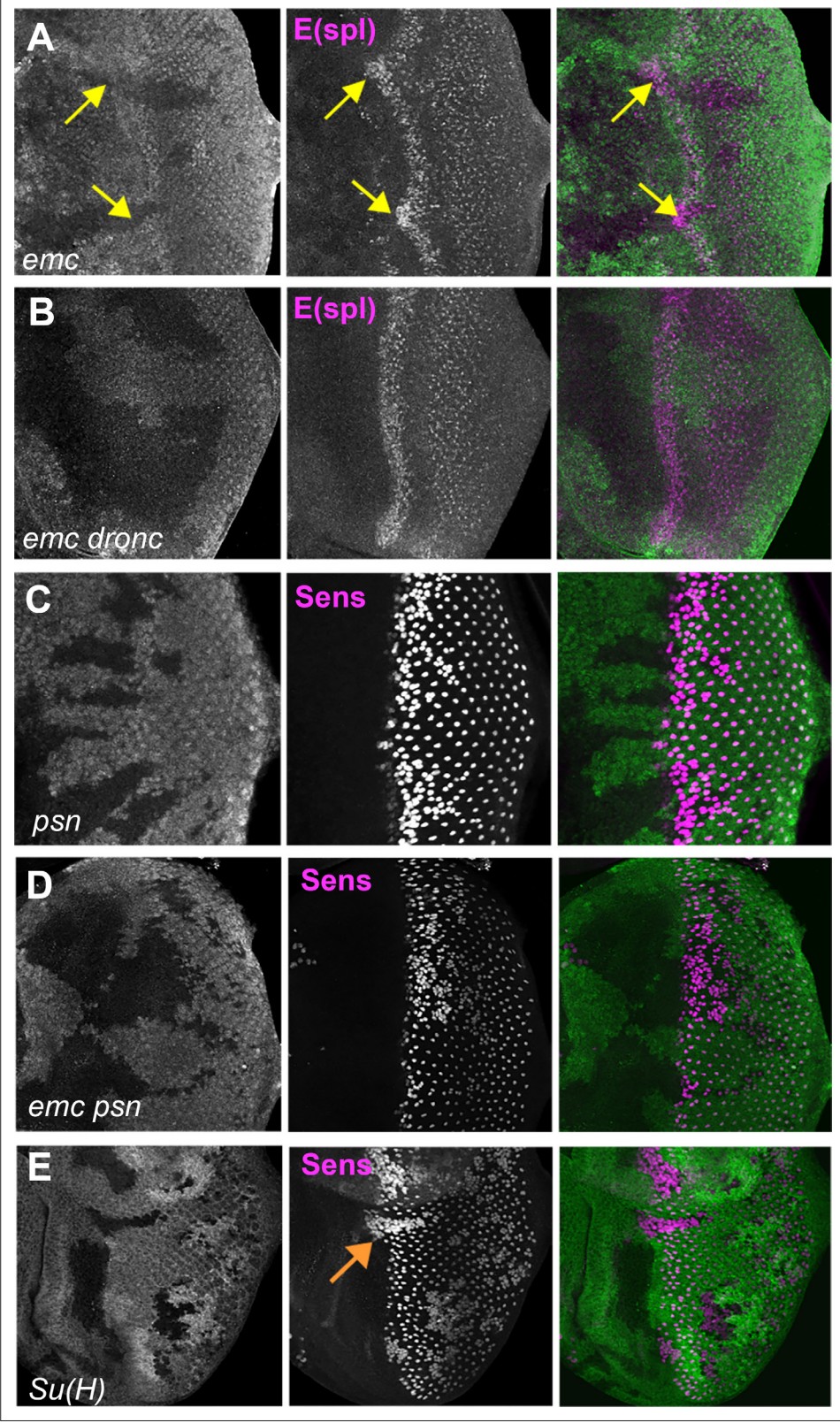

**Figure 6.** Notch activity and function in emc mutant clones. (**A**) E(spl), an important N target for lateral inhibition, is elevated in the morphogenetic furrow region of *emc* clones (yellow arrows). (**B**) *emc dronc* clones have normal levels of E(spl) protein. (**C**) Senseless staining shows a neurogenic phenotype in *psn* mutant clones, due to reduced Notch signaling. Morphogenetic furrow progression is unaffected. (**D**) A neurogenic phenotype was also

*Figure 6 continued on next page*

*Figure 6 continued*

observed in *emc psn* clones, along with normal furrow progression. (**E**) *Su(H)* mutant clones identified by absence of GFP labeling show a strong neurogenic phenotype as well as advanced retinal differentiation (orange arrow) (***Li and Baker, 2001***). The cell-autonomous effect results in a discontinuity at the borders of *Su(H)* clones, where differentiation outside the clones lags that within clones (orange arrow) Genotypes: (**A**) *ywhsF;emc^{AP6} FRT80/ [Ubi-GFP] M(3)67C FRT80*, (**B**) *ywhsF; dronc^{i29}emc^{AP6} FRT80/[UbiGFP] M(3)67C FRT80*, (**C**) *ywhsF/+;psn^{V1} FRT80/ [Ubi-GFP]M(3)67CFRT80*, (**D**) *ywhsF/+; emc^{AP6} psn^{V1} FRT80/[Ubi-GFP]M(3)67CFRT80*, (**E**) *ywhsF; Su(H)^{D47}FRT40/ FRT40[UbiGFP]*. N = 6 for each genotype.

One other known effect of *emc* is that it restrains the *Drosophila* E protein Da. Da is expressed ubiquitously and is the obligate heterodimer partner of proneural bHLH proteins in neurogenesis. In undifferentiated progenitor cells, most or all Da is thought to be sequestered in inactive heterodimers with Emc (***Li and Baker, 2018***). In *emc* mutant cells, Da levels rise and become functional, potentially as homodimers (***Bhattacharya and Baker, 2011***). Multiple aspects of *emc* mutants, including poor growth, speed of morphogenetic furrow progression, and R7 photoreceptor and cone cells fates, depend on this ectopic *da* activity (***Bhattacharya and Baker, 2011***; ***Reddy Onteddu et al., 2024***).

It was noticed, in the context of imaginal disc cell growth, that *emc* mutant cells experience *da*-dependent over-expression of the *ex* gene (***Wang and Baker, 2015b***; ***Wang and Baker, 2018***). Although most study of *ex* has focused on its role in growth control through the SWH pathway, *ex* was shown to influence patterning through Yki-dependent transcription of *diap1*, an important regulator of cell death pathways. This was in the developing thorax, where *ex* mutations affect bristle patterning through *diap1* and a non-apoptotic effect of caspases on Wg signaling (***Wang and Baker, 2019***). This observation led us to explore whether aspects of the *emc* phenotype could reflect reduced *diap1* expression and elevated caspase activity.

Remarkably, we found that all four features of the *emc* phenotype we examined depended on the caspase-mediated cell death pathway. The normal growth of imaginal disc cells, normal speed of the morphogenetic furrow, specification of R7 photoreceptor cells, and timing of non-neuronal cone cell development were all restored when caspase activation was prevented, which was achieved by inactivating the initiator caspase Dronc, deleting the proapoptotic *rpr*, *grim*, and *hid* genes, over-expressing *diap1*, or over-expressing baculovirus p35. These appeared to be non-apoptotic effects because no elevation in apoptosis was apparent in *emc* mutant cells, because generating apoptosis by another means did not mimic the effects, and because the effects were mediated by elevate Dl protein expression in cells that were not apoptotic. They appeared to be effects of caspase activity because they depended in part on *dronc*, an initiator caspase that cleaves and activates effector caspases, because they were suppressed by baculovirus p35, a caspase inhibitor that is a caspase pseudo-substrate, and because they were suppressed by Diap1, a ubiquitin ligase that targets caspases for degradation. The effects may depend on the full caspase cascade because sensitivity to p35 indicates that caspases other than Dronc are necessary, and because mutating *dronc* alone gave lesser degrees of rescue than deleting *rpr*, *grim*, and *hid*. This is consistent with many other studies that describe the pathways for apoptotic and non-apoptotic caspase functions as similar (***Aram et al., 2017***; ***Baena-Lopez et al., 2018***; ***Colon-Plaza and Su, 2022***; ***Kanuka et al., 2005***; ***Kuranaga and Miura, 2007***; ***Nakajima and Kuranaga, 2017***; ***Su, 2020***).

Whereas non-apoptotic caspase activity promotes Wg signaling during specification of thoracic bristles (***Kanuka et al., 2005***), in the *Drosophila* eye the non-apoptotic caspase target was Notch signaling, due to elevated Delta protein levels. We do not know whether Delta is the direct caspase target. Dl protein levels could also be elevated through other mechanisms, for example through elevated *Dl* gene transcription, or through proteins that affect Dl protein stability.

Notably, *emc* mutations accelerate the morphogenetic furrow by enhancing Notch signaling, potentially non-autonomously, but inhibit R7 and cone cell differentiation through cell-autonomously reduced N signaling (***Bhattacharya and Baker, 2009***; ***Reddy Onteddu et al., 2024***). These contrasts may be explained through the dual functions of the Notch ligand Delta in trans-activation and in cis-inhibition (***Bray, 2016***; ***Jacobsen et al., 1998***; ***Li and Baker, 2004***; ***Micchelli et al., 1997***). The eye disc region ahead of the morphogenetic furrow lacks Delta expression (***Baker and Yu, 1998***; ***Parks et al., 1995***). Ectopic Dl expression here is sufficient to activate N non-autonomously and drive morphogenetic furrow progression (***Baonza and Freeman, 2001***; ***Li and Baker, 2001***). Within the

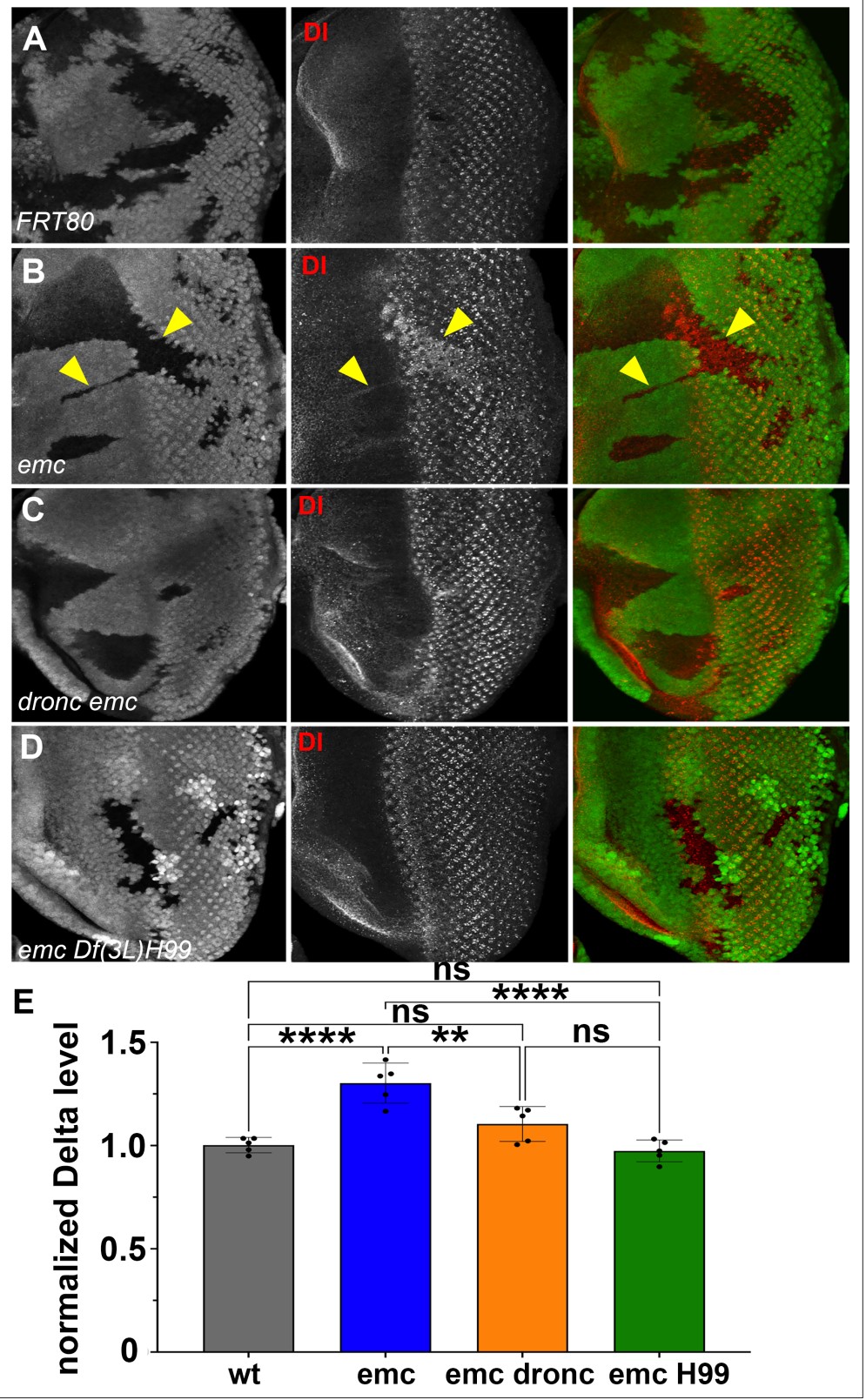

**Figure 7.** Caspases regulate Delta levels. (**A**) Normal levels of Delta protein are seen in FRT80 clones as compared to elevated Delta levels in (**B**) *emc* clones (arrowheads) whereas that phenotype is reversed in (**D**) *emc H99* clones. However, as seen in (**C**), some *dronc emc* clones show intermediate Delta levels, although most resemble wild type. (**E**) Quantification of Delta levels in different clone genotypes, compared to the background

*Figure 7 continued on next page*

*Figure 7 continued*

levels outside the clones. Means ± SEM are shown. Significance was determined using one-way Anova with Tukey's post hoc test. (**p ≤ 0.01, ****p ≤ 0.0001). Source data for (**E**) are provided in *Figure 7—source data 1*. Genotypes: (**A**) *ywhsF;FRT80/[UbiGFP] M(3)67C FRT80*, (**B**) *ywhsF;emc^{AP6}FRT80/[Ubi-GFP] M(3)67C FRT80*, (**C**) *ywhsF;dronc^{i29}emc^{AP6} FRT80/[UbiGFP] M(3)67C FRT80*, (**D**) *ywhsF;emc^{AP6} Df(3L)H99 FRT80/[UbiGFP]FRT80*. N = 7 for each genotype.

The online version of this article includes the following source data and figure supplement(s) for figure 7:

**Source data 1.** Anti-Dl labeling source data for *Figure 7E*.

**Figure supplement 1.** Potential caspase sites predicted in the Delta protein.

R7 equivalence group, Delta is expressed early in R1 and R6 photoreceptor precursors, protecting them from N activation through cis-inhibition, and establishing the distinction between R1,6 and R7 precursor cells (*Miller et al., 2009*). Ectopic Dl expression posterior to the morphogenetic furrow has cis-inhibitory effect. Elevated Dl levels within *emc* mutant clones may render potential R7 and cone cell precursors cell-autonomously resistant to N activation, as normally seen in R1/6 precursor cells.

A previous study suggested that *emc* mutations might affect Hh signaling (*Spratford and Kumar, 2013*). Our data point much more clearly to an effect on Notch signaling. Although the Hh target Ci155 accumulates in *emc* mutant cells, increased Hh signaling is not the only possible cause. Ci155 is cytoplasmic, and thought to be only a precursor for a labile nuclear activator molecule (*Ohlmeyer and Kalderon, 1998*). Preventing activation and nuclear translocation of Ci155 by mutating the kinase *fused* also leads to Ci155 accumulation, but inactivates Hh signaling, showing that Ci155 levels and Hh activity can be separated. Presumably Ci155 accumulates in *fu* mutants because it is more stable than its activated derivative (*Ohlmeyer and Kalderon, 1998*). Significantly, *emc* mutant cells have been reported to over-express the *fused* antagonist *su(fu)*, providing a potential alternative explanation of Ci155 accumulation in *emc* mutant cells (*Spratford and Kumar, 2013*). Analyzing *emc ci* double mutant cells suggested that *ci* was not essential to accelerate the morphogenetic furrow in *emc* mutant clones.

Here, we show that multiple effects of *emc* mutations that occur independently of proneural bHLH genes, due to the loss of restraint on *da* function in *emc* mutant cells, are caused by caspase-dependent non-apoptotic processes that result from reduced expression of Diap1 protein (*Figure 9*). Caspase-dependent non-apoptotic processes have previously been shown to affect Wg activity in the developing notum, although the direct target remains to be identified. Here we show that in eye development, Notch signaling is the target. Expression of Dl protein is increased in a caspase-dependent manner, which leads either to Notch activation or to Notch inhibition, depending on whether Delta acts through trans-activation or cis-inhibition. Our results show that, in addition to contributing to normal development, non-apoptotic caspase activities contribute to mutant phenotypes. In the *Drosophila* eye, multiple aspects of the *emc* phenotype result from caspase-dependent changes in Notch signaling, one of the major cell–cell signaling pathways that contributes to many, many cell fate decisions, and other, unidentified caspase-dependent events affect the growth of *emc* mutant cells in undifferentiated imaginal discs.

Da protein binds to and potentially regulates hundreds of genes throughout the *Drosophila* genome (*Li et al., 2008*). Accordingly, Da activity in *emc* mutant cells might be expected to lead to non-specific and pleiotropic effects. It is therefore remarkable that multiple aspects of proneural bHLH-independent *emc* mutant phenotypes have a simple common basis in elevated caspase activity.

# Materials and methods

## Key resources table

| Reagent type (species) or resource | Designation | Source or reference | Identifiers | Additional information |
|---|---|---|---|---|
| Gene (*Drosophila melanogaster*) | emc | GenBank | FBgn0000575 | |
| Genetic reagent (*D. melanogaster*) | emc [AP6] | PMID:7947322 | FBal0051626 | |

*Continued on next page*

*Continued*

| Reagent type (species) or resource | Designation | Source or reference | Identifiers | Additional information |
|---|---|---|---|---|
| Genetic reagent (*D. melanogaster*) | dronc [i29] | PMID:15800001 | FBal0190283 | |
| Genetic reagent (*D. melanogaster*) | Df(3L)H99 | PMID:8171319 | FBab0022359 | |
| Genetic reagent (*D. melanogaster*) | nkd[3] | PMID:2081466 | FBal0013025 | |
| Genetic reagent (*D. melanogaster*) | PBac{y+ w + ci+}VK33 | This paper | | |
| Genetic reagent (*D. melanogaster*) | psn[v1] | Bloomington *Drosophila* Stock Center | FBal0316340 BDSC: 63237 | |
| Genetic reagent (*D. melanogaster*) | Fz3-RFP | PMID:21869817 | | |
| Genetic reagent (*D. melanogaster*) | pie[eB3] | PMID:1634999 | FBal0032439 | |
| Genetic reagent (*D. melanogaster*) | Su(H)Δ47 | PMID:1617730 | FBal0103950 | |
| Antibody | Emc (Rabbit polyclonal) | Y.N. Jan | | (1:8000) |
| Antibody | Da (Mouse monoclonal) | PMID:3802198 | | (1:200) |
| Antibody | GFP (Rat monoclonal) | Nacalai Tesque | Cat #GF090R RRID:AB_2314545 | (1:50) |
| Antibody | GFP (Rabbit polyclonal) | Invitrogen | Cat #A-11122 RRID:AB_221569 | (1:500) |
| Antibody | β-Gal (Mouse monoclonal) | Developmental Studies Hybridoma Bank | Cat #40-1a RRID:AB_528100 | (1:100) |
| Antibody | β-Gal (Rabbit polyclonal) | Cappel (MP Biomedicals) | Cat #55976 RRID:AB_2313707 | (1:100) |
| Antibody | Runt (Guinea pig polyclonal) | PMID:9683745 | | (1:500) |
| Antibody | E(spl)bHLH mAb323 (Mouse monoclonal) | PMID:1618155 | | (1:50) |
| Antibody | Senseless (Guinea pig polyclonal) | PMID:10975525 | | (1:500) |
| Antibody | Cleaved *Drosophila* Dcp-1 (Rabbit polyclonal) | Cell Signaling Technology | Cat #9578 RRID:AB_2721060 | (1:100) |
| Antibody | DIAP1 (Rabbit polyclonal) | PMID:12021769 | | (1:50) |
| Antibody | phospho-Smad1/5 (Rabbit monoclonal) | Cell Signaling Technology | Cat #9516 RRID:AB_491015 | (1:100) |
| Antibody | Delta (Mouse monoclonal) | Developmental Studies Hybridoma Bank | Cat #C594.9B RRID:AB_528194 | (1:2000) |
| Antibody | Cut (Mouse monoclonal) | Developmental Studies Hybridoma Bank | Cat #2B10 RRID:AB_528186 | (1:50) |
| Antibody | Ptc (Mouse monoclonal) | Developmental Studies Hybridoma Bank | Cat #Apa 1 RRID:AB_528441 | (1:40) |
| Antibody | Elav (Rat monoclonal) | Developmental Studies Hybridoma Bank | Cat #7E8A10 RRID:AB_528218 | (1:50) |
| Antibody | Elav (Mouse monoclonal) | Developmental Studies Hybridoma Bank | Cat #9F8A9 RRID:AB_528217 | (1:100) |

*Continued*

| Reagent type (species) or resource | Designation | Source or reference | Identifiers | Additional information |
|---|---|---|---|---|
| Antibody | Ci (Rat monoclonal) | Developmental Studies Hybridoma Bank | Cat #2A1 RRID:AB_2109711 | (1:10) |
| Antibody | Cy2, Cy3, and Cy5 | Jackson ImmunoResearch | | (1:200) |
| Antibody | Alexa 555 (Guinea pig polyclonal) | Invitrogen | Cat #A-21435 RRID:AB_2535856 | (1:500) |
| Recombinant DNA reagent | genomic Ci | PMID:33084577 | | |
| Commercial assay or kit | ApopTag Red in situ apoptosis detection kit | Millipore Sigma | Cat #S7165 | |
| Software | Cascleave 2.0 | PMID:24149049 | | |

## *Drosophila* strains

The following stocks were employed in this study and were maintained at 25°C unless otherwise stated – *hsflp;emc$^{AP6}$ FRT80/TM6B, hsflp;dronc$^{i29}$FRT80/TM6B, hsflp;dronc$^{i29}$emc$^{AP6}$, hsflp;Df(3L) H99/TM3, hsflp;Df(3L)H99 emc $^{AP6}$/TM3, Ubi-GFP M(3)67C FRT80, FRT42 M(2)56F Ubi-GFP, FRT82 M(3)95A Ubi-GFP, hsflp; Ubi-GFPFRT80, hsflp; Su(H)$^{\Delta 47}$FRT40, pie$^{eB3}$ FRT40, psn[v1]FRT80/TM6B, Fz3-RFP*;D/TM6B (kind gift from Yu Kimata, University of Cambridge). We obtained genomic Ci construct from Kalderon lab and used BestGene Inc to target the transgene to the third chromosome and then recombined these flies to generate *hsflp;ci+M(3)67Calz FRT80/TM6B;ci94/y+spa*.

## Mosaic analysis

Mosaic clones were obtained using FLP/FRT-mediated mitotic recombination (*Golic, 1991*; *Xu and Rubin, 1993*). For non-Minute genotypes, larvae were subjected to heat shock for 30 minutes at 37°C, 60 ± 12 hr after egg laying. For Minute genotypes, heat shock was performed 84 ± 12 hr after egg laying for 50 minutes. Larvae were dissected 72 hr after heat shock. All flies were maintained at 25°C unless otherwise stated.

## Clonal growth measurements

Clone and twin-spot areas were measured by tracing in ImageJ. To quantify the growth effects of various genotypes, the sum of clone areas per wing disc was divided by the sum of twin-spot areas in the same wing disc. This avoids any subjectivity in identifying individual clones and assigning them to individual twin spots. Clone/twin-spot ratios were log-transformed to ensure normality before statistical analysis.

## Immunohistochemistry and histology

Unless otherwise noted, preparation of eye and wing imaginal discs for immunostaining and confocal imaging were performed as described previously (*Baker et al., 2014*). Antibodies from Developmental Studies Hybridoma Bank (DSHB): anti-Ptc (mouse, 1:40), anti-Elav (mouse, 1:100), anti-Elav (rat, 1:50), anti-Cut (mouse, 1:50), anti-Delta C594.9B (mouse, 1:2000), anti-βGal (mouse, 1:100), mouse anti-βGal (1:100, DSHB 40-1a), and anti-Ci (rat, 1:10). Other antibodies: anti-phospho-Smad1/5 (rabbit, 1:100, Cell Signaling), anti-DIAP1 (rabbit, 1:50) (gift from Hyun Don Ryoo, NYU), anti-Dcp1 (rabbit, 1:100, Cell Signaling), anti-Sens (guinea pig, a gift from Hugo Bellen used at 1:500), anti-Da (mouse, 1:200), rabbit anti-Emc (1:8000), anti-GFP (rat, 1:50 from Nacalai Tesque # GF090R), rabbit anti-GFP (1:500), rabbit anti-β-Galactosidase (1:100, Cappel), E(spl)bHLH (1:50,mAb323), and guinea pig anti-runt (1:500). Secondary antibodies conjugated with Cy2, Cy3, and Cy5 dyes (1:200) were from Jackson ImmunoResearch Laboratories and Alexa 555 (1:500). Multi-labelling images were sequentially scanned with Leica SP8 confocal microscopes and were projected and processed with ImageJ. All images were assembled into figure format using Adobe Illustrator 2020.

## Quantifying immunofluorescence

Anti-DIAP1 and anti-Dl labeling were quantified within clones using the average density measurement in ImageJ and normalized to control regions in the same tissue. For anti-DIAP1 labeling, in

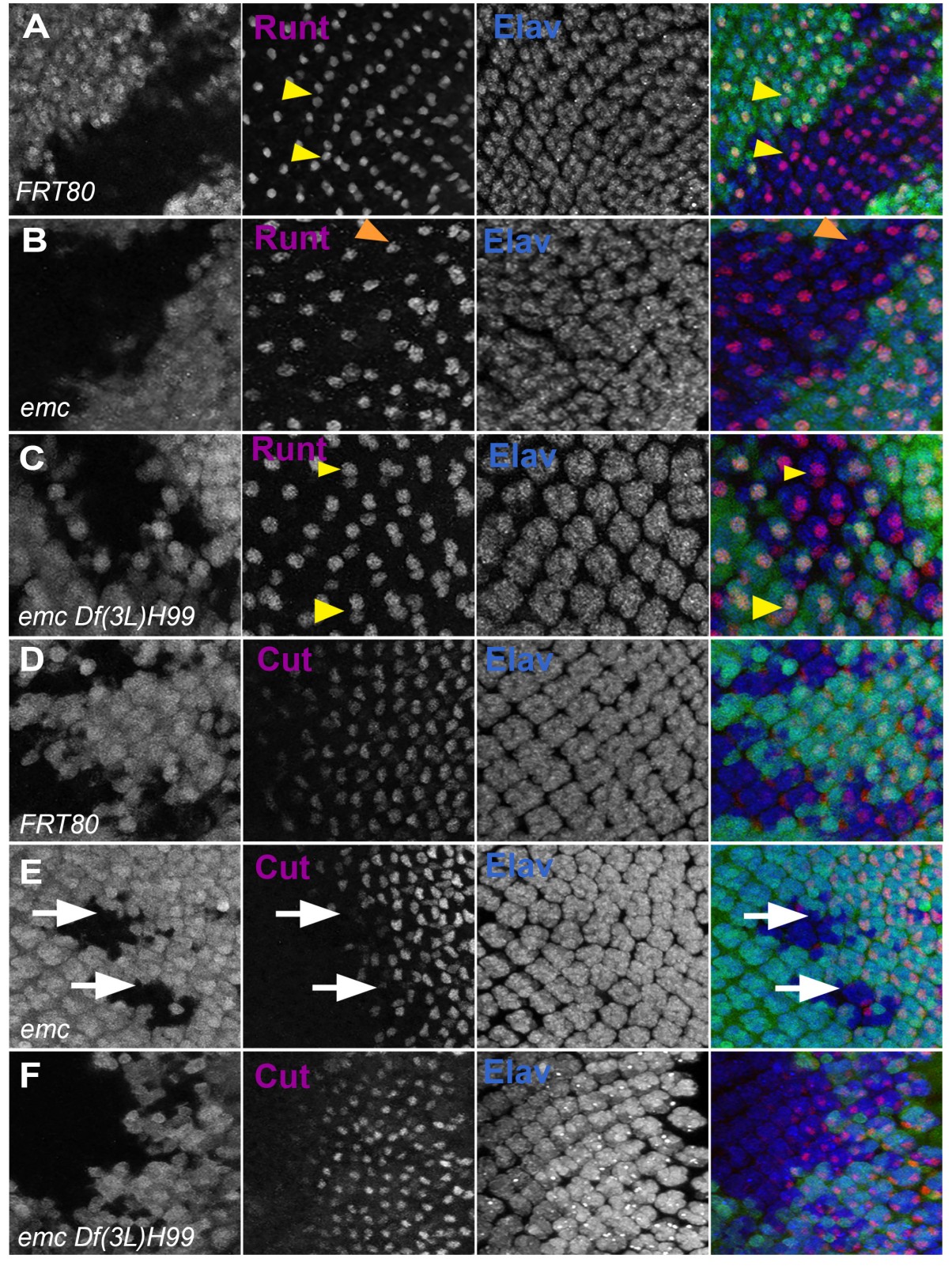

**Figure 8.** Caspases contribute to R7 photoreceptor and cone cell defects in *emc* mutants. In all panels, *emc* mutant cells are marked by the absence of GFP expression (in green) and photoreceptor neurons are marked by Elav in blue. (**A**) Runt (in red) is expressed in R7 and R8 (yellow arrowhead) photoreceptor cells inside and outside the clone in FRT80 controls. (**B**) Inside *emc* clone, Runt expression is lost from R7 cells, while expression in R8 cells remains unaffected (orange arrowhead). (**C**) However, inside the *emc H99* clones, Runt is expressed in both R7 and R8 cells. (**D**) Cut (in red) is

*Figure 8 continued on next page*

*Figure 8 continued*

expressed in cone cells in FRT80 controls. (**E**) Inside *emc* clones, cut is delayed. (**F**) However, inside the *emc H99* clones, cut staining is not delayed. Genotypes: (**A, D**) *ywhsF;FRT80/[UbiGFP] M(3)67C FRT80*, (**B, E**) *ywhsF;emcᴬᴾ⁶FRT80/[Ubi-GFP] M(3)67C FRT80*, (**C, F**) *ywhsF;emcᴬᴾ⁶ Df(3L)H99 FRT80/ [UbiGFP] M(3)67C FRT80. N = 4* for each genotype.

particular, we lack any measurement of any non-specific background labeling that may occur. Differences in Diap1 levels between genotypes may be underestimated if a component of the labeling is non-specific.

### TUNEL assay

For labeling dead cells with TUNEL assay, ApopTag Red In Situ Apoptosis Detection Kit (Cat #S7165) was used according to the manufacturer's instruction. Briefly, dissected eye discs were fixed for 20 min at room temperature followed by three washes with 1× phosphate-buffered saline (PBS). Then the samples were incubated in equilibration buffer for 1 min followed by incubation in reaction buffer (TdT enzyme; ratio 7:3) at 37°C for 1 hr. TdT reaction mix was replaced with stop buffer (diluted 1:34 in dH₂O) and incubated for 10 min at room temperature. Samples were washed three times with 1× PBS, 5 min per wash; and incubated with anti-digoxigenin antibody solution (diluted 31:34 in blocking solution) for 30 min at room temperature. The samples were then washed three times in 1× PBS, 5 min per wash. For the subsequent antibody staining, the samples were blocked in PBST (1× PBS + 0.5% Triton-X) for 30 min, and incubated with primary antibodies in PBST overnight at 4°C. The samples were next washed with PBST and incubated for 2 hr with secondary antibodies in PBST, and then again washed with PBST, followed by PBS wash and samples were mounted in mounting media.

### Statistical analysis

Statistical analysis was performed using GraphPad Prism 7. The statistical tests used are described in the figure legends. Statistical significance is shown as follows: n.s., $p > 0.05$; *$p < 0.05$; **$p < 0.01$; ***$p < 0.001$; ****$p < 0.0001$.

### Prediction of caspase cleavage sites

Caspase cleavage sites were predicted for Delta using Cascleave 2.0 (*Wang et al., 2014*). Cascleave 2.0 was set to a medium stringency threshold for prediction of cleavage sites.

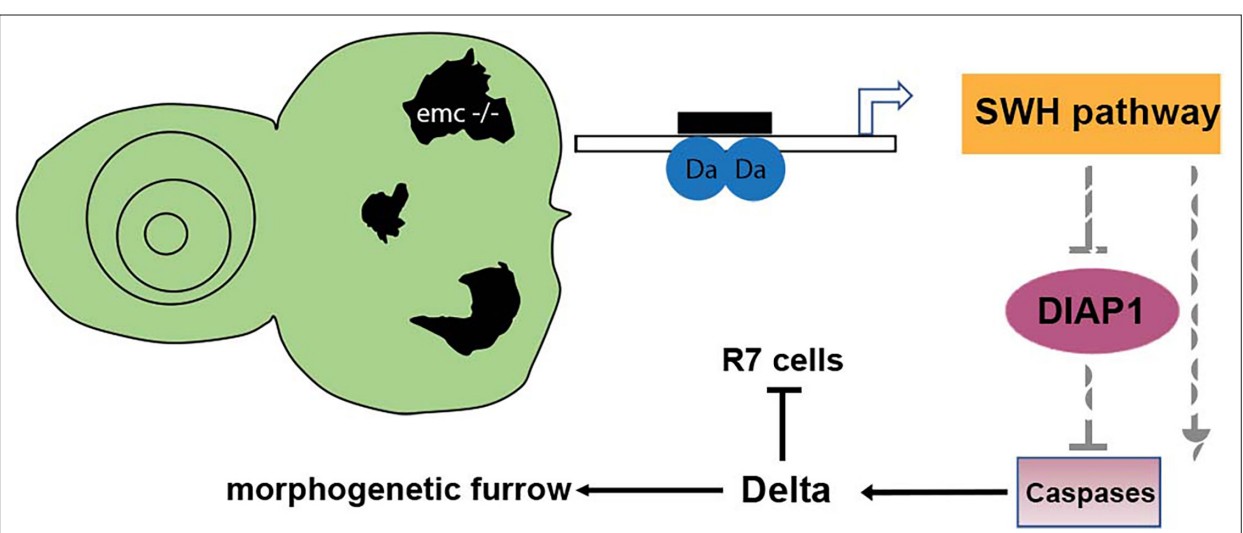

**Figure 9.** Model of *emc* effects on *Drosophila* eye development. Loss of *emc* allows Da protein to form homodimers and activate *ex* transcription, increasing Salvador–Warts–Hippo (SWH) pathway activity. SWH activity reduces DIAP1 expression, thereby derepressing caspase activity. In the eye, non-apoptotic caspase activity increases expression of the Notch ligand Delta. Elevated Delta expression accelerates morphogenetic furrow progression, while cis-inhibiting Notch signaling posterior to the morphogenetic furrow, inhibiting R7 cell specification and cone cell differentiation. In wild-type cells, most Da is likely heterodimerized with either a proneural protein or with Emc protein, and there is no role of caspases in Dl expression.

## Materials availability

All new materials generated in this project are available from the corresponding author.

## Acknowledgements

We thank Y Kimata, D Kalderon, H Bellen, HD Ryoo, and Developmental Studies Hybridoma Bank for antibodies and *Drosophila* strains, and Tao Wang for statistical advice. We thank C Khan, A Kumar, J Secombe, and A Jenny for comments on the manuscript. Confocal microscopy was performed at the Analytical Imaging Facility, Albert Einstein College of Medicine. The Leica SP8 microscope was acquired through NIH SIG 1S10 OD023591. Supported by grants from the NIH (GM047892 and EY028990).

## Additional information

### Funding

| Funder | Grant reference number | Author |
|---|---|---|
| National Institutes of Health | GM047892 | Nicholas E Baker |
| National Institutes of Health | EY028990 | Nicholas E Baker |

The funders had no role in study design, data collection, and interpretation, or the decision to submit the work for publication.

### Author contributions

Sudershana Nair, Conceptualization, Data curation, Formal analysis, Validation, Investigation, Visualization, Methodology, Writing – original draft, Writing – review and editing; Nicholas E Baker, Conceptualization, Data curation, Formal analysis, Supervision, Funding acquisition, Methodology, Project administration, Writing – review and editing

### Author ORCIDs

Nicholas E Baker ⓘ https://orcid.org/0000-0002-4250-3488

Reviewer #1 (Public review): https://doi.org/10.7554/eLife.91988.3.sa1
Reviewer #2 (Public review): https://doi.org/10.7554/eLife.91988.3.sa2
Reviewer #3 (Public review): https://doi.org/10.7554/eLife.91988.3.sa3
Author response https://doi.org/10.7554/eLife.91988.3.sa4

## Additional files

### Supplementary files

MDAR checklist

### Data availability

All data generated or analyzed during this study are included in the manuscript and supporting files; source data files have been provided for *Figures 1, 3 and 7*.

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
