## [Editor Report · eLife Assessment]

This **important** work presents data showing that all non-proneural phenotypes of the Inhibitor of DNA binding (Id) protein Emc are mediated through inappropriate nonapoptotic caspase activity. Using the developing *Drosophila* retina as a model the authors show that Emc acts by transcriptionally regulating the *Death-Associated Inhibitor of Apoptosis 1* (*diap1*) gene, which impacts on Notch signaling by caspase-dependent increase of Delta protein. These are **compelling** findings, interesting for the caspase/apoptosis field as they add more non-apoptotic functions of caspases to the list, as well as for the Id field, which examines how Id proteins inhibit cell differentiation.

---

## [Referee Report · Reviewer #1 (Public review)]

Summary:

The extra macrochaetae (emc) gene encodes the only Inhibitor of DNA binding protein (Id protein) in *Drosophila*. Its best-known function is to inhibit proneural genes during development. However, the emc mutants also display non-proneural phenotypes. In this manuscript, the authors examined four non-proneural phenotypes of the emc mutants and reported that they are all caused by inappropriate non-apoptotic caspase activity. These non-neuronal phenotypes are: reduced growth of imaginal discs, increased speed of the morphogenetic furrow, and failure to specify R7 photoreceptor neurons and cone cells during eye development. Double mutants between emc and either H99 (which deletes the three pro-apoptotic genes reaper, grim, and hid) or the initiator caspase dronc suppress these mutant phenotypes of emc suggesting that the cell death pathway and caspase activity are mediating these emc phenotypes. In previous work, the authors have shown that emc mutations elevate the expression of ex which activates the SHW pathway (aka the Hippo pathway). One known function of the SHW pathway is to inhibit Yorkie which controls the transcription of the inhibitor of apoptosis, Diap1. Consistently, in emc clones the levels of Diap1 protein are reduced which might explain why caspase activity is increased in emc clones giving rise to the four non-neural phenotypes of emc mutants. However, this increased caspase activity is not causing ectopic apoptosis, hence the authors propose that this is non-apoptotic caspase activity. In the last part of the manuscript, the authors ruled out that Wg, Dpp, and Hh signaling are the target of caspases, but instead identified Notch signaling as the target of caspases, specifically the Notch ligand Delta. Protein levels of Delta are increased in emc clones in an H99- and dronc-dependent manner. The authors conclude that caspase-dependent non-apoptotic signaling underlies multiple roles of emc that are independent of proneural bHLH proteins.

Strengths:

Overall, this is an interesting manuscript and the findings are intriguing. It adds to the growing number of non-apoptotic functions of apoptotic proteins and caspases in particular. The manuscript is well written and the data are usually convincingly presented.

Weaknesses:

The authors have addressed all my concerns and questions.

---

## [Referee Report · Reviewer #2 (Public review)]

Id proteins are thought to function by binding and antagonizing basic helix-loop-helix (bHLH) transcription factors but new findings demonstrate roles for emc including in tissues where no proneural (*Drosophila* bHLH) genes are known to function. The authors propose a new mechanism for developmental regulation that entails restraining new/novel non-apoptotic functions of apoptotic caspases.

Specifically, the data suggest that loss of emc leads to reduced expression of diap1 and increased apoptotic caspase activity, which does not induce apoptosis but elevates Delta expression to increase N activity and cause developmental defects. Indeed, many of the phenotypes of emc mutant clones can be rescued by a chromosomal deficiency that reduces caspase activation or by mutations in the initiator caspase Dronc. A related manuscript that shows that loss of emc results in increased da, linked previously to diap1 expression, provides supporting data. There is increasing appreciation that apoptotic caspases have non-apoptotic roles. This study adds to the emerging field and should be of interest to the readers.

The revised manuscript addresses my concerns from the first round of review.

---

## [Referee Report · Reviewer #3 (Public review)]

The work extends earlier studies on the *Drosophila* Id protein EMC to uncover a potential pathway that explains several tissue-scale developmental abnormalities in emc mutants. It also describes a non-apoptotic role for caspases in cell biology.

Strengths:

The work adds to an emerging new set of functions for caspases beyond their canonical roles as cell death mediators. This novelty is a major strength as well as its reliance on genetic-based in vivo study. The study will be of interest to those who are curious about caspases in general.

Weaknesses:

The authors did an adequate job in dealing with the limitations of the reviewed preprint. Although they could have done more, they chose not to for reasons they adequately defended.

---

## [Author Response]

The following is the authors’ response to the original reviews.

**Reviewer #1 (Public Review):**
Summary:The extra macrochaetae (emc) gene encodes the only Inhibitor of DNA binding protein (Id protein) in *Drosophila*. Its best-known function is to inhibit proneural genes during development. However, the emc mutants also display nonproneural phenotypes. In this manuscript, the authors examined four non-proneural phenotypes of the emc mutants and reported that they are all caused by inappropriate non-apoptotic caspase activity. These non-neuronal phenotypes are: reduced growth of imaginal discs, increased speed of the morphogenetic furrow, and failure to specify R7 photoreceptor neurons and cone cells during eye development. Double mutants between emc and either H99 (which deletes the three pro-apoptotic genes reaper, grim, and hid) or the initiator caspase dronc suppress these mutant phenotypes of emc suggesting that the cell death pathway and caspase activity are mediating these emc phenotypes. In previous work, the authors have shown that emc mutations elevate the expression of ex which activates the SHW pathway (aka the Hippo pathway). One known function of the SHW pathway is to inhibit Yorkie which controls the transcription of the inhibitor of apoptosis, Diap1. Consistently, in emc clones the levels of Diap1 protein are reduced which might explain why caspase activity is increased in emc clones giving rise to the four non-neural phenotypes of emc mutants.However, this increased caspase activity is not causing ectopic apoptosis, hence the authors propose that this is nonapoptotic caspase activity. In the last part of the manuscript, the authors ruled out that Wg, Dpp, and Hh signaling are the target of caspases, but instead identified Notch signaling as the target of caspases, specifically the Notch ligand Delta. Protein levels of Delta are increased in emc clones in an H99- and dronc-dependent manner. The authors conclude that caspase-dependent non-apoptotic signaling underlies multiple roles of emc that are independent of proneural bHLH proteins.Strengths:Overall, this is an interesting manuscript and the findings are intriguing. It adds to the growing number of non-apoptotic functions of apoptotic proteins and caspases in particular. The manuscript is well written and the data are usually convincingly presented.Weaknesses:(1) One major concern I have is the observation by the authors in Figure 3C in which protein levels of Diap1 are still reduced in emc H99 double mutant clones. If Diap1 is still reduced in these clones, shouldn't caspases still be derepressed? Given that emc H99 double mutants rescue all emc phenotypes examined, the observation that Diap1 levels are still reduced in emc H99 clones is inconsistent with the authors' model. The authors need to address this inconsistency.

The effect of H99 emc clones on Diap1 protein levels is consistent with our conclusions. The reviewer’s concern probably relates to previous work that shows that RHG proteins act by antagonizing DIAP1, so that Diap1 is epistatic to RHG (PMID:10481910), and that RHG proteins affect DIAP1 protein levels, and in particular that HID promotes DIAP1 ubiquitylation leading to its destruction (PMID:12021767). First, epistasis means that in the absence of DIAP1, RHG levels do not affect cell survival. DIAP1 protein is not absent in emc/emc eye clones, however, it is reduced. It is not only possible but expected that RHG levels would affect survival when DIAP1 levels are only reduced. Secondly, we did not see a difference in DIAP1 levels between H99/H99 clones and H99/+ cells within the same specimen, suggesting that rpr, grim and hid might not affect DIAP1 levels. It is possible that Hid protein only affects DIAP1 levels when overexpressed, as in the aforementioned paper (PMID:12021767), and that physiological RHG levels affect DIAP1 activity. The H99 deficiency also eliminates Rpr and Grim, which may affect DIAP1 without ubiquitylating it. In our experiments, however, there are no cells completely wild type for the H99 region for comparison in the same specimen, so our results do not rule out the H99 deletion having a dominant effect on DIAP1 levels both inside and outside the clones. What our data clearly showed is that emc affected DIAP1 levels independently of any potential RHG effect, and we hypothesized this was through *diap1* transcription, because we showed previously that emc affects yki, a transcriptional regulator of the *diap1* gene, but we have not demonstrated transcriptional regulation of *diap1* directly in *emc* clones. We modified the manuscript to better delineate these issues (lines 275-284).

(2) Are Diap1 protein levels reduced in all emc clones, including clones anterior to the furrow? This is difficult to see in Figure 3B. it is also recommended to look in emc mosaic wing discs.

We now mention that DIAP1 levels were only reduced in *emc* clones posterior to the morphogenetic furrow, not anterior to the morphogenetic furrow or in *emc* clones in wing imaginal discs (lines 284-5) and Figure 3 supplement 1.

(3) The authors speculate that Delta may be a direct target of caspase cleavage (Figure 9B), but then rule it out for a good reason. However, I assume that the increased protein levels of Delta in emc clones (Figure 7) are the results of increased transcription. In that case, shouldn't caspases control the transcriptional machinery leading to Delta expression?

Thank you for suggesting that caspases control the transcription of Dl. We added this possibility to the manuscript (lines 499-500). At one time there was a Dl-LacZ transcriptional reporter, which would have made it straightforward to assess Dl transcription in *emc* clones, but this strain does not seem to exist now. We have not attempted in situ hybridization to *Dl* transcripts in mosaic discs.

(4) How does caspase activity in emc clones cause reduced growth? Is this also mediated through Delta signaling?

We do not know what is the caspase target responsible for reduced growth in wing discs.

(5) Figure 1M: Is there a similar result with emc dronc mosaics?

The *emc dronc* clones do not show as dramatic a growth advantage in a Minute background. This is consistent with the smaller effect of emc dronc in the non-Minute background also (Figure 1N). We mention this in the revised paper (lines 232-3).

**Reviewer #2 (Public Review):**
Id proteins are thought to function by binding and antagonizing basic helix-loop-helix (bHLH) transcription factors but new findings demonstrate roles for emc including in tissues where no proneural (*Drosophila* bHLH) genes are known to function. The authors propose a new mechanism for developmental regulation that entails restraining new/novel non-apoptotic functions of apoptotic caspases.Specifically, the data suggest that loss of emc leads to reduced expression of diap1 and increased apoptotic caspase activity, which does not induce apoptosis but elevates Delta expression to increase N activity and cause developmental defects. Indeed, many of the phenotypes of emc mutant clones can be rescued by a chromosomal deficiency that reduces caspase activation or by mutations in the initiator caspase Dronc. A related manuscript that shows that loss of emc results in increased da, linked previously to diap1 expression, provides supporting data. There is increasing appreciation that apoptotic caspases have non-apoptotic roles. This study adds to the emerging field and should be of interest to readers.The data, for the most part, support the conclusions but I do have concerns about some of the data and the interpretations that should be addressed.
**Reviewer #3 (Public Review):**
The work extends earlier studies on the *Drosophila* Id protein EMC to uncover a potential pathway that explains several tissue-scale developmental abnormalities in emc mutants. It also describes a non-apoptotic role for caspases in cell biology.Strengths:The work adds to an emerging new set of functions for caspases beyond their canonical roles as cell death mediators. This novelty is a major strength as well as its reliance on genetic-based in vivo study. The study will be of interest to those who are curious about caspases in general.Weaknesses:The manuscript relies on imaging experiments using genetic mosaic imaginal discs. It is for the most part a qualitative analysis, showing representative samples with a small number of mutant clones in each. Although the senior author has a long track record of using experiments like this to rigorously discover regulatory mechanisms in this system, it is straightforward in 2023 to use Fiji and other image analysis tools to measure fluorescence. Such measurements could be done for all replicate clones of a given genotype as well as genetic control sampling. These could be presented in plots that would not only provide quantitative and statistical measurements, but will be more reader- friendly to those who are not fly people.

We added quantification of anti-Delta and anti-Diap1 levels to the manuscript (Figures 3E and 7E). We agree that this facilitates statistical confirmation of the results and may be more accessible to non-experts. We do have concerns that these quantifications might be given too much weight. For example, we cannot measure the background level of anti-DIAP1 labeling by labeling *diap1* null mutant cells, because such cells do not survive. Although we measure ~20% reduction in *emc* clones in the eye disc, and none in the wing disc, both measures could be underestimates if some of the labeling is non-specific, as is very possible. We discuss this in the Methods (lines 166-9).

Likewise, more details are needed to describe how clone areas were measured in Figure 1. Did they measure each clone and its twin spot, and then calculate the area ratio for each clone and its paired twin spot? This would be the correct way to analyze the data, yielding many independent measurements of the ratio. And doing so would obviate the need to log transform the data which is inexplicable unless they were averaging clones and twins within a disc and making replicates. More explanation is needed and if they indeed averaged, then they need to calculate the ratios pairwise for each clone and twin.

We added details of clone size measurements and analysis to the methods (lines 141-6). Although it might be useful to compare individual clones and corresponding twin spots, the only rigorous way to associate individual clones with individual twin spots, or even to determine what is one clone and what is one twin spot, is to use recombination rates low enough that significantly less than one recombination occurs per disc. This would require many more dissections and we did not do this. We now clarify in the manuscript that the analysis is indeed based on the ratio of total area of clones and twin spots with replicates, and that Log-transformation is to improve the normality of the ratio data suitable for parametric significance testing, not because clones and twin spots were summed from each sample. We consulted with a statistician over this approach.

**Reviewer #1 (Recommendations For The Authors):**
Lines 319/320: "Frizzled-3 RFP expression was not changed in in emc clones (Figure 4A)". This was actually not shown in Fig 4A (in fact this result was not shown at all). Fig 4A shows the result for emc nkd3 which the authors incorrectly assigned to Figure 4B (line 324).

We apologize for labeling Figure 4A and 4B incorrectly.

The title of Figure 6 is inaccurate. The title does not indicate what is shown in this figure. A more accurate title would be: Notch activity and function in emc mutant clones.

We provided a new title for Figure 6.

**Reviewer #2 (Recommendations For The Authors):**
There is no information on how reproducible the data is. How many discs were examined in each experiment and in how many technical or biological replicates? Can fluorescence signals be quantified within and outside the clones and presented to illustrate reproducibility and significance? This is especially needed for Fig 7, which shows key data that N ligand Delta is elevated in emc clones but dronc and H99 mutations rescue this phenotype. I can see that the Dl signal is brighter in the GFP- emc clone in Fig 7B but I can also see a brighter Dl signal in the small clone and perhaps also in the large clone in C. The difference between B and C could be simply disc-to-disc variation, which should be addressed with quantification and presentation of all data points.

We added the number of samples to each figure legend. We quantified the fluorescence signals for Figures 3 and 7. Quantification shows that the difference between 7B and 7C is highly significant, not disc to disc variation.

Fig 2B does not support the conclusion. It is supposed to show premature Sens expression and therefore abnormal morphogenetic furrow progression in emc clones. But the yellow arrow is pointing to GFP+ (wild type) cells and it is within this GFP+ region that most premature Sens expression is seen.

We relocated the arrows in Figure 2B to point precisely to the premature differentiation. When the morphogenetic furrow is accelerated in emc mutant, GFP – tissue, it does not stop when wild type, GFP+ tissue is encountered again, it continues at a normal pace. Accordingly, emc+ regions that are anterior to emc- regions can also experience accelerated differentiation (please see lines 594-8).

Fig 1 shows that while H99 deficiency restores the growth of emc clones to wild type level (Fig 1N), placing these in the Minute background made emc clones grow better than emc wild type but Minute neighbors (Fig 1M). The latter cells were nearly absent, suggesting elimination through cell competition. For the rest of the figures, some experiments are done in the Minute background (e.g., emc H99 clones in Fig 2D) while others are not in the Minute background (e.g., emc H99 clones in Fig 7D). Why the switch between backgrounds from experiment to experiment?

Figure 2D shows emc H99 clones in a Minute background so that it can be compared with panels 2A-C, which show clones of other genotypes in a Minute background. These clones almost take over the eye disc. In Figure 7D, it was important to show the Dl expression pattern in a substantial wild type region, which could only be shown using the non-Minute background. We have no indication that a Minute background changes the properties of the nonMinute clone, other than allowing its greater growth.

The first 3 paragraphs of the Introduction are overly detailed and read more like a review article. These could be made more concise to focus on the founding data for this manuscript, which are the published findings that emc mutations elevate ex expression (line 129) and that ex mutants show elevated diap1 expression (line 125). These do not show up until the very end of the Introduction.

We shortened the Introduction to focus more rapidly on the topics relevant to these experiments.

In several places, the space between the end of the sentence and the citation is missing (e.g., lines 57, 68, and 75).

The spacing of citations was fixed.

Line 247. 'morphogenetic furrow that found each ommatidia...' should use a word besides 'found.'

We corrected line 247.

**Reviewer #3 (Recommendations For The Authors):**
(1) The authors show that inhibiting caspases rescues the growth defect of emc clones. However, they did not find excessive TUNEL staining in emc clones that would explain why the clones would be so small - excessive cell death. How reliable was their tunel staining in being able to detect excessive apoptosis (only negative data was shown). Could they induce excessive cell death using radiation or some other means to ensure the assay is robust? If death is not occurring in emc clones, a deficiency worth addressing is that they do not discuss or explore how the caspases then inhibit clone growth. Is it expanded cell cycle times, or smaller cells?? And that phenotype does not fit with their end model of Delta being the only moderator of emc since it is not playing a significant role in tissue growth anterior to the furrow.One would assume using the commercial antibody against activated caspase would be another readout for emc clones and this would bolster their claim that excessive caspase activation occurs in the emc cells.

We have added Dcp1 staining in Figure 2 supplement 3 to show that TUNEL staining is reliable.

(2) Figure 3D has really large emc clones when GMR-Diap is present. But the large clones are anterior to the furrow where Diap would not be overexpressed. Is this just an unusual sample with a coincidentally big emc M+ clone? It speaks to my concerns about the qualitative nature of the data.

We replaced Figure 3D with an example of smaller clones. Nowhere have we suggested that GMR-DIAP1 affects clone size.

(3) Figure 9B is very speculative and not appropriate since the authors have zero data to support that cleavage mechanism. It is fit for the next paper if the idea is correct. The panel should be removed.

We did not intend Figure 9B to imply that we think Dl itself is the relevant target of non-apoptotic caspases. Since apparently we gave that impression, we removed this to a supplemental figure. We still think it is worth showing that Dl does not contain predicted caspase sites expected to activate signaling.

(4) Figure 9A could be made more clear. Their pathway represents the mutant cells in the mosaic disc. Why not also outline what you think is happening in the emc+ cells as well?

It is difficult to make a comparable diagram for normal cells, because none of this pathway happens in normal cells. We modified the figure legend to indicate this (lines 677-8).

(5) The one emc ci clone they show spanning the furrow has a very non-continuous furrow advance phenotype. This is unlike the emc clones where the furrow advance is graded about the clone. And it resembles the SuH clones they show. This result and the synergistic effect on clone sizes they mention need more discussion and thought put into it. It argues ci is doing something with respect to emc action. loss of ci might not rescue size and furrow advance but actually, it makes it worse! This is interesting and might suggest an inhibitory role for ci in emc or a parallel role for ci in mediating growth and progression that is redundant with emc.

We agree that aspects of the emc ci phenotype are not clear. We discuss this in the revised manuscript (lines 373-5).

(6) Related to point 7, it is a weak argument for non-autonomy that graded furrow advance in emc clones is evidence for emc acting nonautonomously through Delta. Its weakness is combined with its lack of significance relative to the other findings. It should be deleted as should the SuH data.

We agree that the evidence that emc affects morphogenetic furrow progression non-autonomously is not compelling and have revised the manuscript to soften this conclusion (lines 426-7). We do not want to remove this idea, because it does in fact have significance for other findings. Specifically, it supports the idea that the emc effect in the morphogenetic furrow is due to trans-activation by Delta, whereas the effect on R7 and cone cell differentiation is due to autonomous cis-inhibition. We think this is important to keep in the paper.